# Intrinsic dynamic shapes responses to external stimulation in the human brain

Maximilian Nentwich[1]*, Marcin Leszczynski[2,3,4], Charles E Schroeder[2,3], Stephan Bickel[1,3,5], Lucas C Parra[6]*

[1]The Feinstein Institutes for Medical Research, Northwell Health, Manhasset, United States; [2]Departments of Psychiatry and Neurology, Columbia University College of Physicians and Surgeons, New York, United States; [3]Translational Neuroscience Lab Division, Center for Biomedical Imaging and Neuromodulation, Nathan Kline Institute, Orangeburg, United States; [4]Cognitive Science Department, Institute of Philosophy, Jagiellonian University, Kraków, Poland; [5]Departments of Neurology and Neurosurgery, Zucker School of Medicine at Hofstra/Northwell, Hempstead, United States; [6]Department of Biomedical Engineering, The City College of New York, New York, United States

*For correspondence:
max.nentwich@gmail.com (MN);
parra@ccny.cuny.edu (LCP)

Competing interest: The authors declare that no competing interests exist.

## eLife Assessment

This manuscript presents an interesting new framework (VARX) for simultaneously quantifying effective connectivity in brain activity during sensory stimulation and how that brain activity is being driven by that sensory stimulation. The reviewers thought the model was original and its conclusion that intrinsic connectivity is reduced (rather than increased) during sensory stimulation is very interesting, but that for ideal performance, one must specify all sensory features in the model, which is not possible. Overall, however, this work is **important** with **convincing** evidence for its conclusions - it will be of interest to neuroscientists working on brain connectivity and dynamics.

**Abstract** Sensory stimulation of the brain reverberates in its recurrent neural networks. However, current computational models of brain activity do not separate immediate sensory responses from this intrinsic dynamic. We apply a vector-autoregressive model with external input (VARX), combining the concepts of 'functional connectivity' and 'encoding models', to intracranial recordings in humans. This model captures the extrinsic effect of the stimulus and separates that from the intrinsic effect of the recurrent brain dynamic. We find that the intrinsic dynamic enhances and prolongs the neural responses to scene cuts, eye movements, and sounds. Failing to account for these extrinsic inputs leads to spurious recurrent connections that govern the intrinsic dynamic. We also find that the recurrent connectivity during rest is reduced during movie watching. The model shows that an external stimulus can reduce intrinsic noise. It also shows that sensory areas have mostly outward, whereas higher-order brain areas have mostly incoming connections. We conclude that the response to an external audiovisual stimulus can largely be attributed to the intrinsic dynamic of the brain, already observed during rest.

## Introduction

The primate brain is highly interconnected between and within brain areas. This includes areas involved in sensory processing (*Felleman and Van Essen, 1991*). Strikingly, most computational models of brain activity in response to external natural stimuli do not take the recurrent architecture of brain networks

into account. We will refer to the dynamic driven by this recurrent architecture as the *intrinsic dynamic* of the brain. 'Encoding' models often rely on simple input/output relationships such as general linear models in fMRI (*Friston et al., 1995*), or temporal response functions (TRFs) in EEG/MEG (*Lalor and Foxe, 2010*). Interactions between brain areas are captured often just as instantaneous linear correlations that are referred to as 'functional connectivity' when analyzing fMRI activity (*Greicius et al., 2003*). Others capture synchronous activity in different brain areas by measuring phase locking of electrical neural signals (*Varela et al., 2001*). However, these measures of instantaneous correlation do not capture time delays inherent in recurrent connections. By taking temporal precedence into account with recurrent models, the 'Granger-causality' formalism can establish directed 'connectivity'. This has been used to analyze both fMRI and electrical activity (*Friston et al., 2013*; *Haufe et al., 2013*; *Pellegrini et al., 2023*; *Seth et al., 2015*; *Sheikhattar et al., 2018*; *Soleimani et al., 2022*).

The concept of functional connectivity was first developed to analyze neural activity during rest, where there are no obvious external signals to stimulate brain activity. But it is now also used to analyze brain activity during passive exposure to a stimulus, such as watching movies (*Betti et al., 2013*; *Geerligs et al., 2015*; *Mennes et al., 2013*; *Vanderwal et al., 2017*). A general observation of these studies is that a portion of the functional connectivity is preserved between rest and stimulus conditions, while some aspects are altered by the perceptual task (*Betti et al., 2013*; *Demirtaş et al., 2019*), sometimes showing increased connectivity during the stimulus (*Vanderwal et al., 2017*). This should be no surprise, given that an external stimulus can drive multiple brain areas and thus induce correlations between these areas (*Cole et al., 2019*). Removing such stimulus-induced correlations by controlling for a common cause is standard practice in statistical modeling and causal inference (*Pearl, 2013*). However, in studies that focus on functional connectivity in neuroscience, stimulus-induced correlations are often ignored when analyzing the correlation structure of neural signals. A notable exception is 'dynamic causal modeling' (*Friston et al., 2003*). In this modeling approach, the 'input' can modulate functional connectivity. This is particularly important in the context of active behavioral tasks, where the common finding is that correlation structure changes with task states (*Gonzalez-Castillo and Bandettini, 2018*).

In this study, we are interested in 'passive' tasks such as rest and movie watching. We will ask here whether, after removing some of the stimulus-induced correlations, the intrinsic dynamic is similar between stimulus and rest conditions. Attempts to factor out the effects of intrinsic dynamics from that of the stimulus come from work on response variability. For instance, fMRI shows that variability across trials in motor cortex is due to an intrinsic 'noise' which is linearly superimposed on a common response in both hemispheres due to a simple motor action (*Fox et al., 2006*). Stimulus–response variability in the visual cortex has been attributed to variability of the ongoing dynamic (*Arieli et al., 1996*; *Buzsaki, 2006*). Some studies of electrical recordings from the visual cortex show that correlations of spiking activity between different recording locations are largely unaffected by visual stimulation (*Fiser et al., 2004*). Yet, other studies show that visual input affects local correlation in the visual cortex (*Gray et al., 1989*; *Ito et al., 2020*; *Nauhaus et al., 2009*) and across the brain (*Roelfsema et al., 1997*).

The technical challenge when addressing these questions is to separate the direct effect of the stimulus from the intrinsic dynamic. Here, we propose to separate these effects by modeling them simultaneously with the simplest possible model, namely, linear intrinsic effects between brain areas and linear responses to extrinsic input. A mathematical model that implements this is the vector-autoregressive model with external input (VARX). This model is well established in the field of linear systems (*Ljung, 1999*) and econometrics (*Hamilton, 2020*), where it is used to capture intrinsic dynamics in the presence of an external input. The VARX model is an extension of the VAR model that is routinely used to establish 'Granger-causality' in neuroscience (cited above). In the VARX model, Granger analysis provides a measure of statistical significance for the external input, as well as for the intrinsic dynamic, including its directionality, all as part of a single model (*Parra et al., 2025*).

While linear systems are an inadequate model of neuronal dynamics, they remain an important tool to understand neural representations because of their conceptual simplicity. They are routinely used for event-related fMRI analysis but also for 'encoding models' to link nonlinear features of continuous stimuli to neural responses. They have been used to analyze responses to video in fMRI (*Naselaris et al., 2011*), to speech in EEG (*Di Liberto et al., 2015*) or to audio in intracranial EEG (*Holdgraf et al., 2017*). They are even used to analyze the encoding in deep-neural network models (*Li et al.,*

*2023*). Here, we use a classic linear model to combine two canonical concepts in neuroscience, which have thus far remained separated, namely, that of 'encoding models' (*Naselaris et al., 2011*) and 'functional connectivity' models (*Friston et al., 2013*). We will use this to analyze whole-brain, intracranial EEG in human patients at rest and while they watch videos. Our main finding is that the recurrent connections observed during rest are only minimally altered by watching videos. Instead, the brain's response to naturalistic stimulus appears to be substantially shaped by the same intrinsic dynamic of the brain observed during rest.

## Results

### Extrinsic input leads to spurious recurrent connectivity

To determine the effect of the extrinsic inputs on connectivity estimates, we either fit a VARX model or a VAR model (i.e. a VARX model with no external input). We analyze LFP data on all available recordings, movies, and resting state for all $N = 26$ recording sessions. As extrinsic inputs, we included film cuts, motion, fixation onset, fixation novelty, the sound envelope, and acoustic edges. VAR models contain the same external inputs as the VARX model, but the time alignment is disrupted by a circular shuffle. This keeps the number of parameters in different models constant

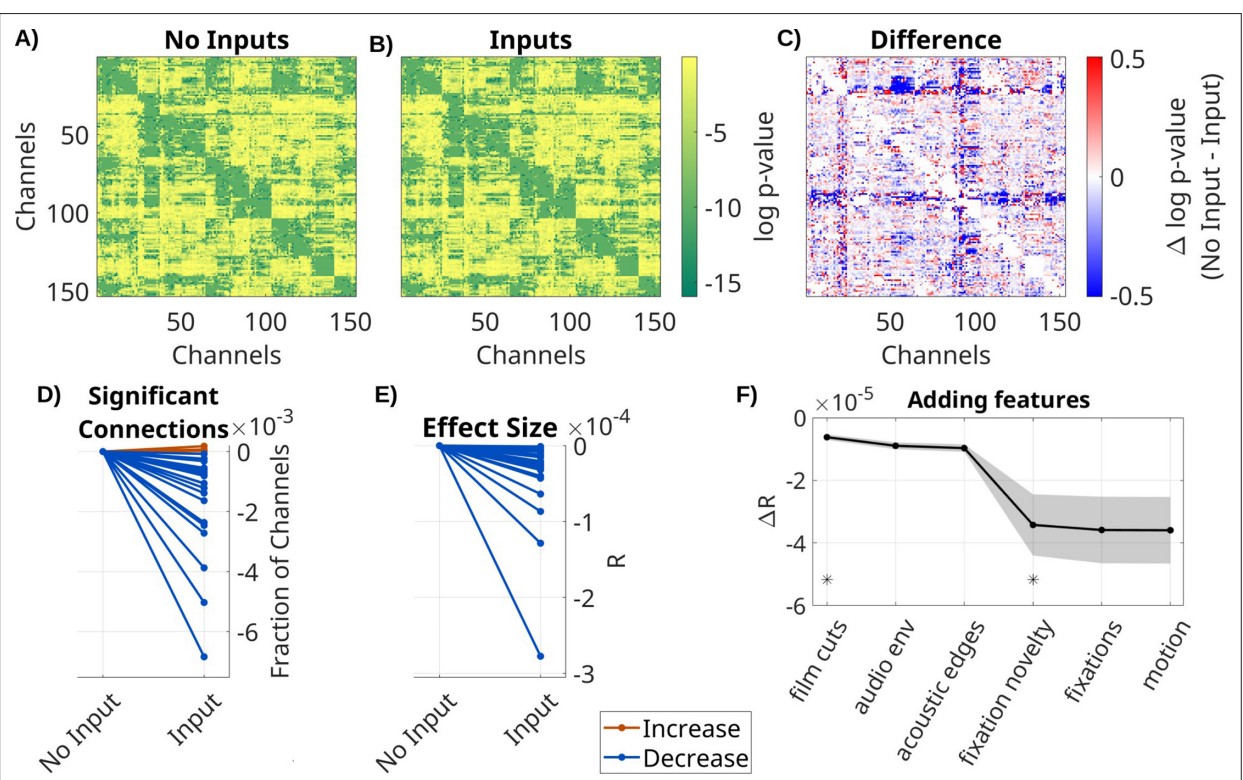

**Figure 1.** Spurious recurrent connectivity in *A* is removed when modeling the effect of extrinsic input with *B*. Comparison of VARX model with and without inputs. (**A**) log *p*-values for each connection in *A* for a VARX model without inputs on one patient (Pat_1); (**B**) for a VARX model with inputs; (**C**) difference of log *p*-values for VARX model without minus with inputs (panels A and B). Both models are fit to the same data. (**D**) Thresholding panels A and B at p < 0.0001 gives a fraction of significant connections. Here, we show the fraction of significant channels for models with and without input. Each line is a patient with color indicating increase or decrease. (**E**) Mean over all channels for VARX models with and without inputs. Values in (**D**) and (**E**) have been normalized to models without input. (**F**) Change in *R* values when successively adding inputs to the VARX model. Black line shows mean across patients, shaded gray area the standard error of the mean. Stars indicate features that further reduce effect size over the previously added feature with statistical significance (Wilcoxon rank sum test p < 0.05). Negative values indicate a decrease in connectivity strength when the extrinsic inputs are accounted for. Results for broadband high-frequency activity (BHA) are shown in *Figure 1—figure supplement 2*.

The online version of this article includes the following figure supplement(s) for figure 1:

**Figure supplement 1.** Effect of individual extrinsic features on effect size of recurrent connections.

**Figure supplement 2.** Spurious recurrent broadband high-frequency activity (BHA) connectivity in *A* is accounted for when modeling the effect of input with *B*.

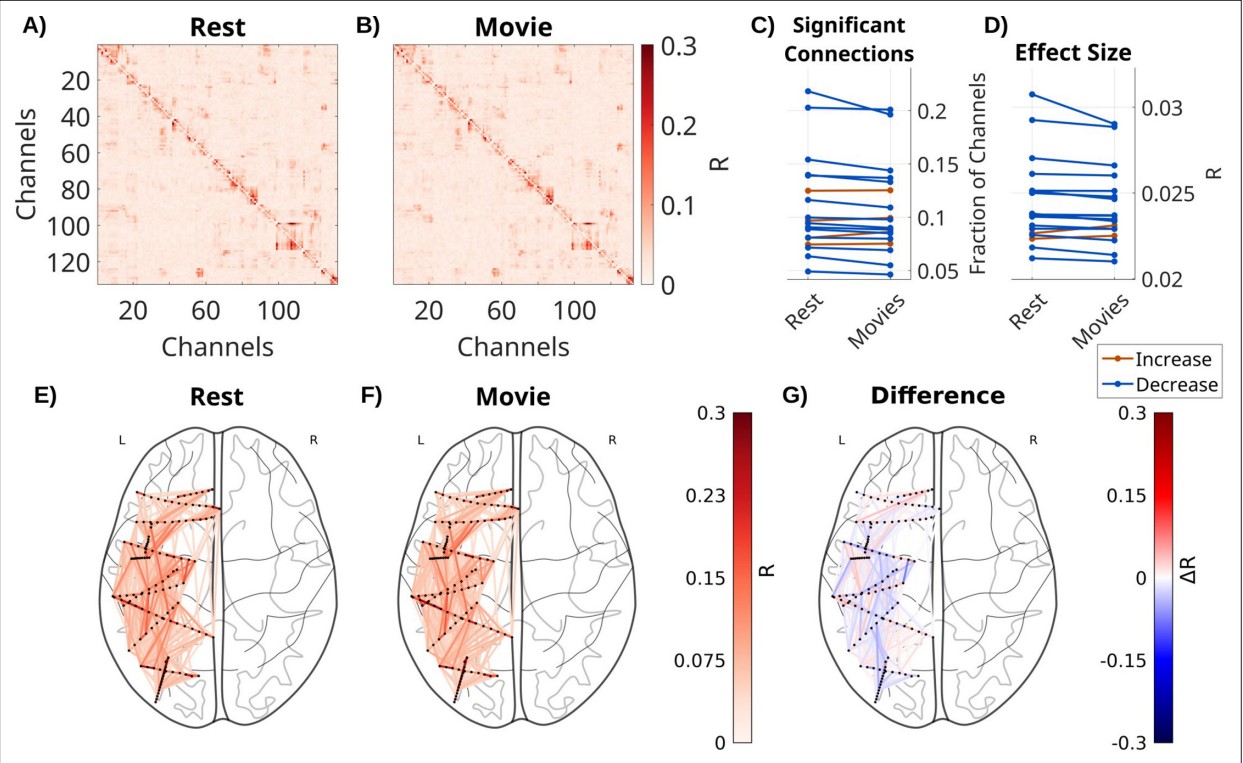

**Figure 2.** Recurrent connectivity during movies is decreased compared to rest. Effect size *R* for each connection in **A** for one patient (Pat_7) for (**A**) a VARX model during resting fixation with fixation onset as input feature. (**B**) The same with a VARX model of local field potential (LFP) recordings during movie watching, with the following input features: sound envelope, acoustic edges, fixation onsets, fixation novelty, motion, and film cuts. (**C**) Fraction of significant connections (p < 0.0001) for movies and rest. (**D**) Mean effect size across all channels for movies and rest. Each line is a patient, with color indicating a numerical increase or decrease. For the movie conditions, we averaged across four different 5-min movie segments. (**E**) Axial view of significant connections in resting state. Black dots show the location of contacts in MNI space. Lines show significant connections between contacts (p < 0.001) colored in red according to effect size *R*. For plotting purposes, connections in the upper triangle are plotted, and asymmetries are ignored. (**F**) The same for the movie task, and (**G**) the difference between movies and resting state, showing both increases and decreases for specific connections. Differences between broadband high-frequency activity (BHA) recurrent connectivity **A** during movies and resting state are shown in *Figure 2—figure supplement 2*.

The online version of this article includes the following figure supplement(s) for figure 2:

**Figure supplement 1.** Recurrent connectivity decreases in movies compared to eyes-closed rest.

**Figure supplement 2.** Recurrent connectivity *A* of broadband high-frequency activity (BHA) decreases during movie watching compared to rest.

**Figure supplement 3.** Eye movement behavior differs between movies and resting state.

and ensures the inputs have the same covariance structure. We found a similar connectivity structure for the estimated VAR and VARX models (*Figure 1A, B*). However, they vary systematically in the number of significant recurrent connections **A** (those with p < 0.0001, *Figure 1D*), which drops when adding inputs (median = $-7.3 \times 10^{-4}$, p < 0.0001, N = 26, Wilcoxon). The effect sizes *R* also significantly decrease in the VARX model (*Figure 1E*, median = $-2.2 \times 10^{-5}$, p < 0.0001, N = 26, Wilcoxon). Therefore, accounting for the external input removes spurious 'connections'. We also analyzed how much each of these extrinsic inputs contributed to this effect. Removing any of the input features increased the effect size of recurrent connections compared to a model with all features (*Figure 1—figure supplement 1*). We then cumulatively added each feature to the VARX model. Effect size monotonically decreases with each feature added (*Figure 1F*). Decreases of effect size are significant when adding film cuts ($\Delta R = -3.6 \times 10^{-6}$, p < 0.0001, N = 26, FDR correction, $a = 0.05$) and the sound envelope ($\Delta R = -3.59 \times 10^{-6}$, p = 0.002, N = 26, FDR correction, $a = 0.05$). Thus, adding more input features progressively reduces the strength of recurrent 'connections'.

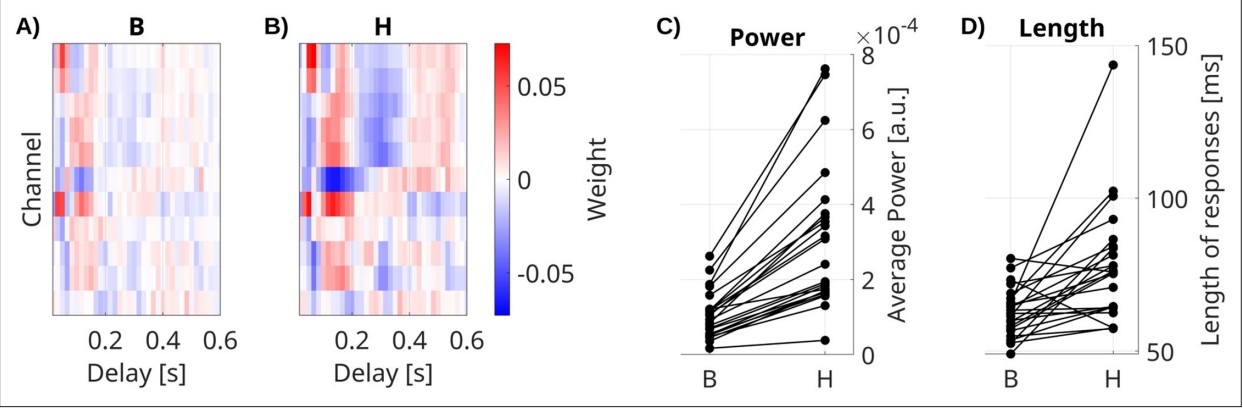

**Figure 3.** Impulse response models. (**A**) Feed-forward responses B to fixation onset are weaker and shorter than (**B**) the overall system response H. Impulse response models for fixation onset in channels with significant responses for one example patient. (**C**) Power and (**D**) mean length of responses in significant channels for all patients. Each line is a patient. Responses to fixation onset in all significant channels, as well as auditory envelope and film cuts, are shown in *Figure 3—figure supplement 2*.

The online version of this article includes the following figure supplement(s) for figure 3:

**Figure supplement 1.** Broadband high-frequency activity (BHA) impulse response models.

**Figure supplement 2.** Impulse response models.

**Figure supplement 3.** Robustness of estimates of input filters *B* to intrinsic effects and uncorrelated features.

## Recurrent connectivity is reduced during movies compared to rest

Next, we compared recurrent connectivity between movie watching and rest (*Figure 2*). In the rest condition, patients have a fixation cross on a gray background. This obviously reduces the size and number of saccades as compared to movie watching, but does not abolish them (*Figure 2—figure supplement 3*). We therefore use a VARX model including fixation onset as an extrinsic variable in both cases. Movies include fixation novelty, film cuts, the sound envelope, acoustic edges, and motion as external inputs. To control for the number of free parameters, we include copies of features from the movies in the resting state model. The number of significant recurrent connections in ***A*** was significantly reduced during movie watching compared to rest (*Figure 2C*, fixed effect of stimulus: beta = $-3.8 \times 10^{-3}$, $t(88) = -3.9$, $p < 0.001$), as is the effect size (*Figure 2D*, fixed effect of stimulus: beta = $-2.5 \times 10^{-4}$, $t(88) = -4.1$, $p < 0.001$). While the effect size decreases on average, there is some variation across different brain areas (*Figure 2E–G*). In a subset of patients with eyes-closed resting state, we find the same effect, which is qualitatively more pronounced (*Figure 2—figure supplement 1*).

## Recurrent connectivity enhances and prolongs stimulus responses

We also compared the feed-forward extrinsic effect ***B*** with the total system response ***H***, which includes the additional effect of the recurrent connectivity ***A***. We estimate ***B*** with the VARX model (*Figure 3A*) on data during video watching and resting state and estimate the total response ***H*** directly using TRFs (*Figure 3B*). Both models include fixation onset, film cuts, and sound envelope as external inputs. We compare the power and length of filters from both models (*Figure 3C, D*). We compare responses in channels with significant effects of ***B*** (FDR correction, $a = 0.05$). We see that the total response ***H*** of fixation onset is significantly stronger (*Figure 3C*, median$\Delta$ = $1.5 \times 10^{-4}$, $p < 0.0001$, $N = 23$, Wilcoxon) and longer than the feed-forward effect ***B*** (*Figure 3D*, median$\Delta$ = 10.9 ms, $p = 0.0002$, $N = 23$, Wilcoxon). The same effect is observed for BHA (*Figure 3—figure supplement 1*) and other input features (*Figure 3—figure supplement 2*). This suggests that the total response of the brain to these external inputs is dominated by the intrinsic dynamics of the brain.

As with conventional linear regression, the estimate in ***B*** for a particular input and output channel is not affected by which other signals are included in ***x*(t)** or ***y*(t)**, provided those other inputs are uncorrelated. We confirmed this here empirically by removing dimensions from ***y*(t)** (*Figure 3—figure supplement 3A*), and by adding uncorrelated input to ***x*(t)** (*Figure 3—figure supplement 3B*, adding fixation onset does not affect the estimate for auditory envelope responses). In other words, to estimate ***B***, we do not require all possible stimulus features and all brain activity to be measured and

included in the model. In contrast, **B** does vary when correlated inputs are added to **x**(t) (**Figure 3—figure supplement 3B**, adding acoustic edges changes the auditory envelope response). Evidently, the auditory envelope and acoustic edges are tightly coupled in time, whereas fixation onset is not. When a correlated input is missing (acoustic edges), then the other input (auditory envelope) absorbs the correlated variance, thus capturing the combined response of both.

## Results are similar for VARX models of BHA and LFP

We repeated the same analyses of *Figures 1–3* with broadband high-frequency activity (BHA). While local field potentials (LFPs) are thought to capture dendritic currents, BHA is correlated with neuronal firing rates in the vicinity of an electrode. Generally, we find a more sparse recurrent connectivity for BHA as compared to LFP (compare *Figures 1 and 2* with *Figure 1—figure supplement 2* and *Figure 2—figure supplement 2*). Perhaps this is expected, given that LFP covers a broader frequency range. Regardless of this overall difference, we find similar results when analyzing BHA with the VARX model. Namely, taking the extrinsic input into account removed stimulus-induced recurrent connections (*Figure 1—figure supplement 2*); the fraction of significant channels decreases during movie watching compared to rest (*Figure 2—figure supplement 2*); and responses to the stimulus are stronger and more prolonged when separately modeling the effect of recurrent connectivity (*Figure 3—figure supplement 1*). In the Discussion section, we will argue that some of these results are expected in general when decomposing the total system response into extrinsic and intrinsic effects.

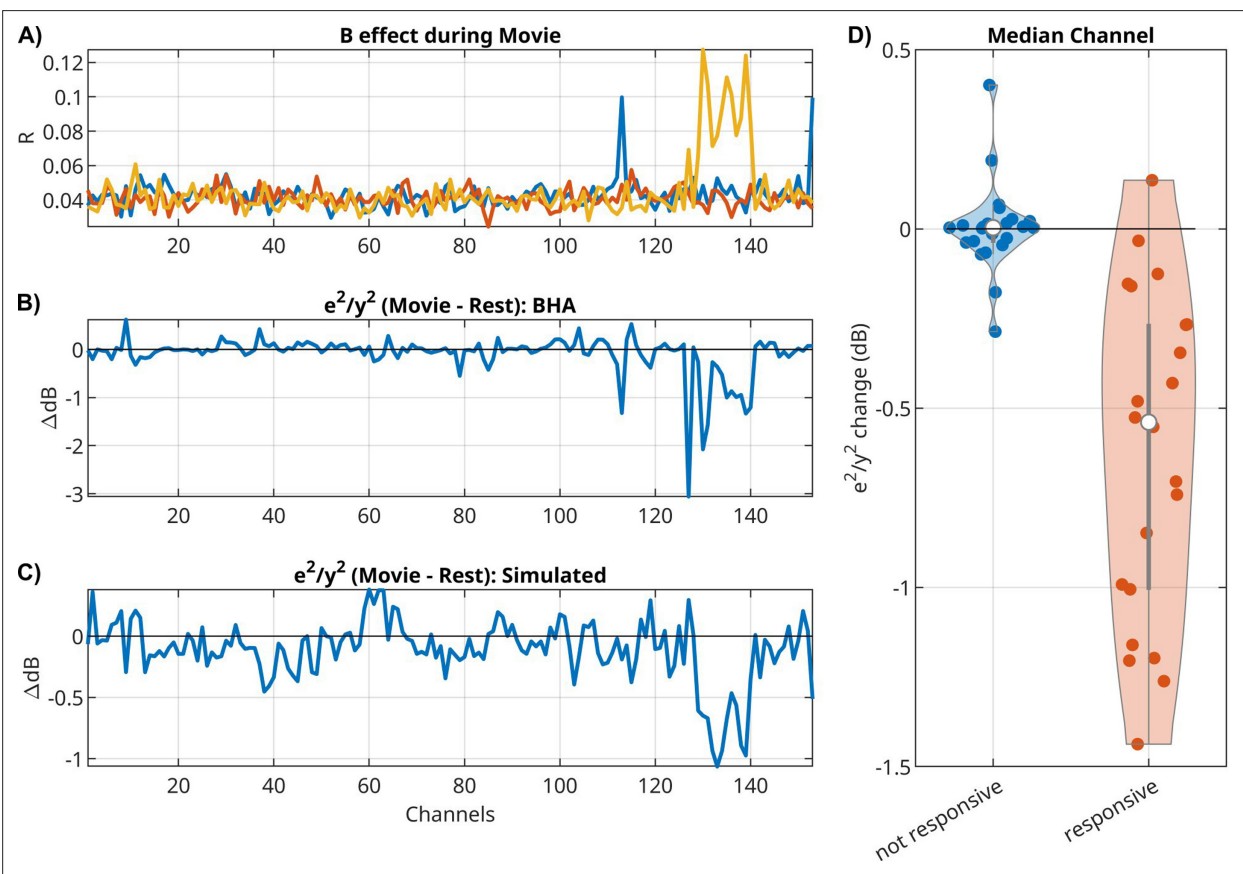

**Figure 4.** For broadband high-frequency activity (BHA), relative power of innovation versus signal drops during movies as compared to rest in responsive channels. (**A**) Effect size $R$ for extrinsic effect **B** in all channels for three input features (scene cuts, fixation onset, and sound envelope). In this example, 15 electrodes had significant responses to one of the three inputs (Bonferroni corrected at p < 0.01). (**B**) Change in relative power of innovation (dB(innovation power/signal power), then subtracting movie − rest). (**C**) Change in relative power of innovation in a simulation of a VARX model with gain adaptation. Here, we are using the **A** and **B** filters that were estimated on BHA on the example from panels A and B. (**D**) Median of power ratio change across all patients, contrasting responsive versus non-responsive channels.

## Intrinsic 'noise' in BHA is reduced by external stimulus

So far, we have discussed the mean response captured by **B** and the activity mediated by **A**. We now want to analyze whether the external input modulates the internal variability of brain activity. As a metric of internal variability, we measured the power of the intrinsic innovation process **e**(t), which captures the unobserved 'random' brain activity that leads to variations in the responses. For the LFP signal, we see a drop in power during movies as compared to rest, for both the original signal **y**(t) (*Appendix 1—figure 1A*) and the model's innovation process **e**(t) (*Appendix 1—figure 1B*). Notable is the stronger oscillatory activity during rest (*Appendix 1—figure 1A*). In this example, we see a drop in power in the theta/alpha band (5–11 Hz) during movie watching across all electrodes (*Appendix 1—figure 1A*, dotted lines). We observe a similar narrow-band drop in power in most patients, albeit at different frequencies (not shown). When analyzing BHA, we find no difference in the power of the innovation process between movie and rest, but we do find a drop in power relative to the overall BHA signals for some channels (*Figure 4B*). These channels seem to coincide with channels that responded to the external stimuli, that is channels with a significant effect in **B** (*Figure 4A*). If we take for each patient the median relative power for responsive channels (median among those with p < 0.0001), then we find that relative power drops for nearly all patients (*Figure 4D*, Wilcoxon rank sum test, p = 8.8 × 10⁻⁷, N = 26). The motivation for analyzing only responsive channels comes from a simple gain adaptation (*Appendix 1—figure 2*). Gain adaptation keeps the power constant, so that the extra power injected by the stimulus implicitly reduces the relative power of the innovation process. This effect is specific to channels receiving external input (*Appendix 1—figure 2D*) and absent in a linear system without gain adaptation (*Appendix 1—figure 2C*). To demonstrate that this simple gain adaptation can explain the noise quenching in the neural data, we simulated data with the gain adaptation model (*Figure 4C*) using parameters estimated for the example patient of *Figure 4A, B*.

## Direction of connectivity differs with cortical hierarchy

Finally, we measured the directionality of the recurrent connections in the LFPs by analyzing the structure of the resulting *R* matrices of all patients, combining data from all available movies and resting state recordings. Columns in *R* represent outgoing connections, while rows are incoming connections. Therefore, the difference of $R-R^T$ (*Figure 5A*) averaged along a column has positive values if a node has overall stronger outgoing connections, and negative values if it has stronger incoming connections. We measured this directionality for each channel across all patients and averaged also across

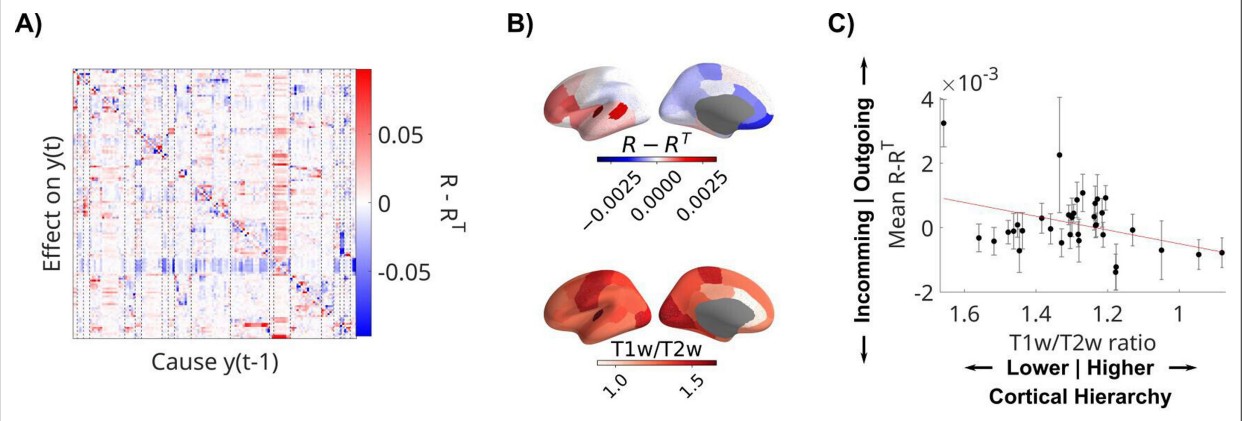

**Figure 5.** Recurrent connectivity of local field potential (LFP) is directed from sensory to higher-order areas. (**A**) Difference of $R-R^T$ showing asymmetric directed effects. Dashed lines indicate regions of interest in the Desikan–Killiany atlas. (**B**) Mean directionality across patients and T1w/T2w ratio are averaged in parcels of the Desikan–Killiany atlas. (**C**) Mean directionality is correlated with cortical hierarchy, estimated with the T1w/T2w ratio. Each dot represents a parcel in the Desikan–Killiany atlas with error bars indicating error of the mean across patients with channels in that parcel. The datapoint on the top left is the transverse temporal gyrus. Note that the *x*-axis has been flipped to show areas higher on the cortical hierarchy on the right. T1w/T2w ratio and cortical hierarchy have an inverse relationship.

The online version of this article includes the following figure supplement(s) for figure 5:

**Figure supplement 1.** Directionality of recurrent broadband high-frequency activity (BHA) connectivity in relation to cortical hierarchy.

channels within parcels of the Desikan–Killiany atlas ($N$ = 34 regions of interest, *Figure 5B*; *Desikan et al., 2006*). We expected this to co-vary with 'cortical hierarchy'. To test this, we compared this asymmetry metric with the T1w/T2w ratio, which captures gray matter myelination and is used as an indirect measure of cortical hierarchy (*Gao et al., 2020a*; *Wang, 2020*). We also average the T1w/T2w ratio in the same parcels of the Desikan–Killiany atlas (*Figure 5B*). We used a mixed-effects model and found that cortical areas showing more outgoing connections ($R$-$R^T$>0) have higher T1w/T2w ratio, which are located lower on the cortical hierarchy ($t$(533) = 2.62, p = 0.009, *Figure 5C*). BHA analysis shows the same effect (*Figure 5—figure supplement 1*).

## Discussion

Our results suggest that the duration and magnitude of responses to extrinsic input are in large part a result of the intrinsic dynamic of the recurrent brain network. We also found that the intrinsic dynamic had reduced recurrent connectivity and weaker intrinsic variability during the movie stimulus.

### Response to extrinsic input versus intrinsic dynamics

Previous literature does often not distinguish between intrinsic dynamics and extrinsic effects. By factoring out some of the linear effects of the external input, we conclude here that recurrent connectivity is reduced on average. From our prior work (*Nentwich et al., 2023*), we know that the stimulus features we included here capture a substantial amount of variance across the brain in intracranial EEG. Arguably, however, the video stimuli had rich semantic information that was not captured by the low-level features used here. Adding such semantic features could have further reduced shared variance, and consequently further reduced average recurrent connectivity in the model.

Similarities and differences between rest and movie watching conditions reported previously do not draw a firm conclusion as to whether overall 'functional connectivity' is increased or reduced. Results seem to depend on the time scale of neural activity analyzed and the specific brain networks (*Betti et al., 2013*; *Demirtaş et al., 2019 Vanderwal et al., 2015*). However, in fMRI, the conclusion seems to be that functional connectivity during movies is stronger than during rest (*Vanderwal et al., 2017*), which likely results from stimulus-induced correlations. The VARX model can remove some of the effects of these stimuli, revealing that average recurrent connectivity may be reduced rather than increased during stimulus processing. Reduced functional connectivity has previously been observed within the visual cortex when a visual stimulus is presented (*Nauhaus et al., 2009*).

In this work, we focused on 'passive' tasks, that is resting with gaze on a fixation point, versus watching movies without any associated tasks. We did not analyze data during an active task requiring behavioral responses. The literature on active tasks emphasizes 'state change' in functional connectivity (*Mennes et al., 2013*; *Gonzalez-Castillo and Bandettini, 2018*; *Cole et al., 2014*). Efforts to factor out task-evoked activity when computing functional connectivity concord with our conclusions that connectivity is inflated by a task (*Cole et al., 2019*). Nevertheless, we hesitate to extrapolate our findings to active tasks, as we have not analyzed such data. Future studies should test if our findings replicate in independent iEEG datasets, including active tasks and whether they generalize to other neuroimaging modalities.

Conventional 'encoding' models, such as TRFs, capture the total response **H** of the brain to an external stimulus. Here, we factored this into a moving average filter **B**, followed by an autoregressive filter **A**. The important observation is that this intrinsic dynamic governed by **A** does not change during stimulus processing. Arguably then, the role of the initial responses **B** is to shape the input to be processed by the existing intrinsic dynamic. This interpretation is consistent with the view of 'the brain from the inside out' advocated by György Buzsáki (*Buzsaki, 2019*). In this view, learning of a stimulus representation consists in learning a mapping of the external stimulus to an existing intrinsic dynamic of the brain.

### Similar findings for LFP and BHA

We found more sparse recurrent connectivity for BHA as compared to LFP. This may be expected because correlations in lower frequencies (that dominate LFPs) reach over longer distances compared to correlations in higher frequencies (e.g. *Muller et al., 2016*). BHA has been linked to a mixture of neuronal firing and dendritic currents (*Leszczyński et al., 2020*), in contrast to LFP, which is thought

to originate from widespread dendritic currents. Despite the observed differences in sparsity, for both LFP and BHA, we found that modeling the intrinsic dynamic removed spurious recurrent connections. Removal of spurious effects when controlling for a common cause is a generic finding in multivariate statistical models. We also found for both LFP and BHA that the duration and strength of stimulus responses can be largely attributed to the recurrent connections. Arguably, this is a generic feature of an autoregressive model, as it more readily captures longer impulse responses. However, the extrinsic filters $B$ in principle have an advantage as they can be fit to each stimulus and brain location. In contrast, the recurrent filters $A$ are constrained by having to capture a shared dynamic for all stimulus dimensions. Thus, the predominance of the intrinsic dynamic in the total system response is not a trivial result of the factorization into intrinsic and extrinsic effects.

## Stimulus-induced reduction of noise in the intrinsic activity

One difference we did find between LFP and BHA is the intrinsic innovation process, that is the internal sources of variability or 'noise'. For both BHA and LFP, we saw a drop in the magnitude of signal fluctuations during the movie watching condition. For the BHA but not the LFP, this was explained as a drop in intrinsic noise. Specifically, for BHA, there was less relative power in the intrinsic 'noise' for channels that are responsive to the stimulus. This is consistent with the notion that response variability is due to variability of intrinsic activity (*Arieli et al., 1996*), which is found to decrease across the brain with the onset of an external stimulus (*Churchland et al., 2010*). This type of noise quenching has been associated with increased attention (*Arazi et al., 2019*) and improved visual discrimination performance (*Arazi et al., 2017*). The effect we found here can be explained by a VARX model with the addition of a divisive gain adaptation mechanism that keeps the total power of brain activity constant. When the input injects additional power, this nonlinear gain adaptation implicitly reduces the contribution of the intrinsic noise to the total power. The noise-quenching result and its explanation via gain adaptation show the benefit of using a parsimonious linear model, which can suggest nonlinear mechanisms as simple corrections from linearity.

We also observed an overall drop in LFP power during movie watching. This phenomenon was strongest in oscillatory bands, with frequencies in theta (5–8 Hz) to beta (15–25 Hz) band differing across patients. In scalp EEG, noise quenching is associated with a similar overall drop in power with the stimulus (*Arazi et al., 2019*). This quenching of neural variability was also found to reduce correlation between brain areas for fMRI and neural spiking (*Ito et al., 2020*). Both fMRI and neural spiking correlated with BHA (*Mukamel et al., 2005*).

## Stimulus features

During the movie and rest periods, we utilized fixation onset to capture activity that is time-locked to visual processing because patients move their eyes even during rest. We also added the fixation novelty regressor to capture semantic changes in the visual input across eye movements (*Nentwich et al., 2023*). We incorporated the sound envelope, a prominent feature known for capturing the dominant audio-induced variance in scalp EEG (*Di Liberto et al., 2015*), as well as acoustic edges that capture strong transients in the auditory input (*Forseth et al., 2020*; *Oganian and Chang, 2019*). In addition, we included film cuts as features, as we had previously demonstrated that they dominate the response in the BHA across the brain (*Nentwich et al., 2023*). Motion is added as an additional feature to capture, while other basic visual features such as overall optic flow or fixations on faces elicited responses in the BHA, their contribution was relatively smaller. The analysis is not limited to these few features, and future research should explore which stimulus features capture variance in the data and how they affect the apparent recurrent connectivity. There is a substantial body of literature on encoding models of semantic features, where nonlinear features of a continuous natural stimulus are extracted and then linearly regressed against fMRI (*Huth et al., 2016*; *Nishimoto et al., 2011*) or EEG (*Broderick et al., 2018*). This work can be directly replicated with the VARX model, which further models the recurrent connectivity.

## Alternative approaches

The traditional VAR model has been used extensively in neuroscience to establish directed 'Granger causal' connections (*Barnett and Seth, 2014*). This approach has been very fruitful and found

numerous extensions (*Sheikhattar et al., 2018*; *Soleimani et al., 2022*). However, these model implementations do not specifically account for an external input.

A few methods have attempted to model the effect of varying task conditions on functional connectivity, mostly in the analysis of fMRI. One approach is to first model the task-evoked responses, equivalent to estimating $B$ alone, and then compute the conventional 'functional connectivity', i.e. the correlation matrix, on the residuals $e(t)$ (*Fair et al., 2007*). Others suggested estimating $B$ in multiple time windows and then estimating 'task-related functional connectivity' by correlating the multiple $B$ over time windows (*Rissman et al., 2004*). It is not clear that these ad hoc methods systematically separate intrinsic from extrinsic factors.

A more principled modeling approach is 'dynamic causal modeling' (DCM) (*Friston et al., 2003*) and extensions thereof (*Ryali et al., 2011*). Similar to the VARX model, DCM includes intrinsic and extrinsic effects $A$ and $B$. However, the modeling is limited to first-order dynamics (i.e. $n_a = n_b = 1$). Thus, prolonged responses have to be entirely captured with a first-order recurrent $A$. In contrast, the DCM includes a multiplicative interaction of extrinsic input $x(t)$ on the connectivity $A$, which does not exist in the VARX model. This interaction has been used to explicitly model a change in recurrent connectivity with task conditions. Here we found that this may not be necessary for intracranial EEG. A practical advantage of the VARX model is the assumption that the neural activity is directly observed. Instead, many existing models assume an error in the observations, which triggers computationally intensive estimation algorithms, typically the expectation maximization algorithm. The same is true for the 'output error' model in linear systems theory (*Ljung, 1999*). As a result, these models are often limited to small networks to test specific alternative hypotheses (*Penny et al., 2004*). (The original DCM proposed for fMRI included an added complication of modeling the hemodynamic response, which amounts to adding a temporal filter to each output node and prior to adding observation noise.) In contrast, here we have analyzed up to 300 channels per patient across the brain, which would be prohibitive with DCM. By analyzing a large number of recordings, we were able to draw more general conclusions about whole-brain activity.

## Caveats

The stimulus features we included in our model capture mostly low-level visual and auditory input. It is possible that regressing out a richer stimulus characterization would have removed additional stimulus-induced correlation. While we do not expect that this would change the overall effect of a reduced number of 'connections' during movie watching compared to resting state, the interpretation of changes in specific connections will be affected by the choice of features. For example, in sensory cortices, higher recurrent connectivity in the LFP during rest would be consistent with the more synchronized state we saw in rest, as reflected by larger oscillatory activity. Synchronization in higher-order cortices, however, is expected to be more strongly influenced by the semantic content of external input.

We find a correlation of diffusion tensor imaging (DTI) structural connectivity used in a model with a VARX estimate of 0.70. That is considered a relatively large value compared to other studies that attempt to recover DTI connectivity from the correlation structure of fMRI activity (*Honey et al., 2009*). A caveat is that this was done on a biophysical model of firing rate, not fMRI, and we have not explored the parameters of the model that might affect the results.

We used fixation onsets as external input, but it should be noted that they are tightly correlated in time with saccade onsets (there is only about a 30 ms jitter between the two, depending on saccade amplitude). While saccades are driven by visual movement, they are generated by the brain itself and arguably could also be seen as intrinsic. The same is true for all motor behaviors, most of which cause a corresponding sensory response, similar to the visual response following a saccade. Including them as external input is a modeling choice we have made here, but it is important to acknowledge that fixation onsets can therefore have 'acausal' components (*Nentwich et al., 2023*). By 'acausal', we mean a fixation-locked response that precedes the fixation onset and is due to the neural activity leading up to the saccade and subsequent fixation. Such acausal responses can be captured by the VARX Granger formalism by delaying the input relative to the neural activity, which we have not done here.

The correlation between the average incoming and outgoing connections and cortical hierarchy (*Figure 5*) is not significant when normalizing for the number of electrodes in each region of interest.

**Table 1.** Models commonly used in neural signal analysis.

(a) 'Interact' refers to an additional bilinear interaction term of the form x C y that allows for a modulation of intrinsic effect by the external input. (b) The DCM is defined in terms of the first derivative of y(t), which in discrete time is the same as $n_a = 1$. (c) It is straightforward to add an interaction term to the VARX model and maintain fast OLS estimation.

| Model | Intrinsic effect A | Extrinsic effect B | Interact | Delay $n_a$, $n_b$ | Estimation speed | Reference, with code where available |
|---|---|---|---|---|---|---|
| GLM | No | Yes | No | = 1 | Medium | *Friston et al., 1994*, SPM, FSL |
| DCM | Yes | Yes | Yes[a] | = 1[b] | Slow | *Friston et al., 2003*, no code |
| VAR | Yes | No | No | >1 | Fast/slow | *Barnett and Seth, 2014* |
| mTRF | No | Yes | No | >1 | Fast | *Crosse et al., 2016* |
| VARX | Yes | Yes | No[c] | >1 | Fast | *Parra et al., 2025* |

Regions in the temporal lobe with a large number of electrodes might drive this correlation. A more fine-grained analysis in these regions could be the goal of future analysis.

## Conclusion

We analyzed whole-brain intracranial recordings in human patients at rest and while they watched videos. We used a model that separates intrinsic dynamics from extrinsic effects. We found that the brain's response to the audiovisual stimuli appears to be substantially shaped by its endogenous dynamics. The model revealed a small but significant decrease in recurrent connectivity when watching movies. Finally, we observed a reduction in intrinsic variance during the extrinsic stimulus, which may be the result of neuronal gain adaptation.

## Materials and methods

The vector-autoregressive model with external input (VARX) falls within a group of well-established linear models used in neuroscience (see *Table 1*). Prominent examples in this group are the generalized linear model (GLM), DCM, and TRF. While these models have been extensively used for neural signal analysis, the VARX model has not. We start, therefore, with a brief introduction. For more details, please refer to *Parra et al., 2025*.

### VARX model

The VARX model explains a time-varying vectorial signal *y(t)* as the result of an intrinsic autoregressive feedback driven by an innovation process *e(t)* and an extrinsic input *x(t)*. (We adopt here the terminology of 'intrinsic' and 'extrinsic' as it is commonly used in neuroscience and psychology. In system modeling and econometrics, where the VARX model is prevalent, the more common terminology is 'endogenous' and 'exogenous' (meaning the same things).) For the *i*th signal channel, the recurrence of the VAX model is given by:

$$y_i(t) = \sum_{j=1}^{d_y} \sum_{\tau=1}^{n_a} A_{ij}(\tau) y_j(t-\tau) + \sum_{j=1}^{d_x} \sum_{\tau=0}^{n_b} B_{ij}(\tau) x_j(t-\tau) + e_i(t)$$

***A*** and ***B*** are matrices of filters of lengths $n_a$ and $n_b$ respectively. Therefore, ***A*** has dimensions and has dimensions [$d_y$, $d_x$, $n_b$], where $d_y$, $d_x$ are the dimensions of ***y****(t)* and ***x****(t)* respectively. The innovation process ***e****(t)* captures the internal variability of the model. Without it, repeating the same input ***x****(t)* would always result in a fixed deterministic output ***y****(t)*. The innovation is assumed to be uncorrelated in time and therefore has a uniform spectrum. The recurrent filters ***A*** modify this spectrum to match the spectrum of ***y****(t)*, thereby capturing the intrinsic dynamic. The feed-forward filters ***B*** inject a filtered version of the extrinsic input ***x****(t)* into this intrinsic dynamic. The role of each of these terms for brain activity is explained in *Figure 6*. We will refer to the filters in the matrix ***A*** and ***B*** as recurrent and feed-forward 'connections', but avoid the use of the word 'causal', which can be misleading (*Parra et al., 2025*).

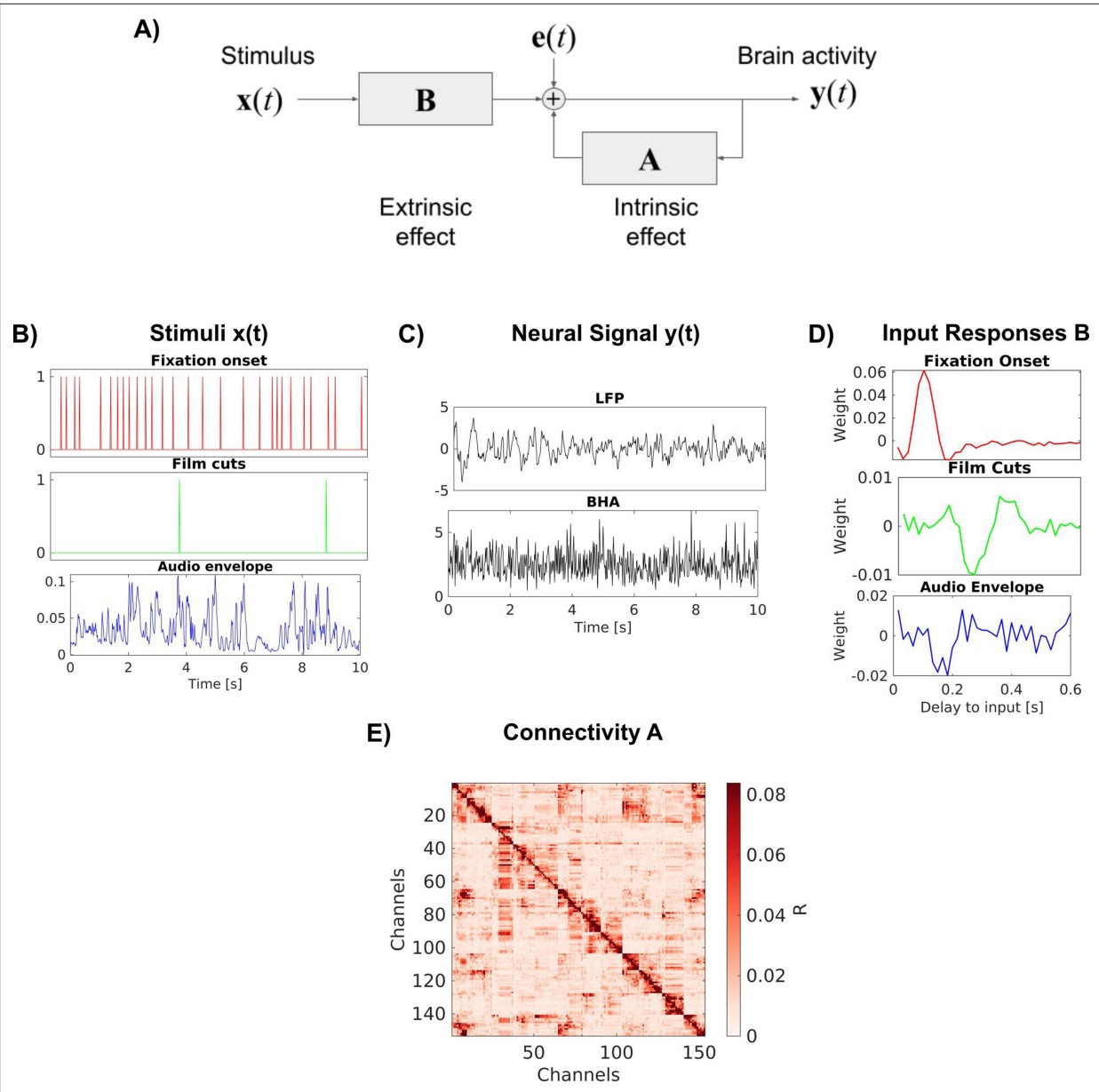

**Figure 6.** VARX model of the brain. (**A**) Block diagram of the VARX model. *y(t)* represents observable neural activity in different brain areas, *x(t)* are observable features of a continuous sensory stimulus, *A* represents the recurrent connections within and between brain areas (intrinsic effect), and B captures the transduction of the sensory stimuli into neural activity and transmission to different brain areas (extrinsic effect). The diagonal term in **A** captures recurrent feedback within a brain area. Finally, *e(t)* captures unobserved 'random' brain activity, which leads to intrinsic variability. (**B**) Example of input stimulus features **x**(*t*). (**C**) Example of neural signal **y**(*t*) recorded at a single location in the brain. We analyze local field potentials (LFPs) and broad-band high-frequency activity (BHA) in separate analyses. (**D**) Examples of filters **B** for individual feed-forward connections between an extrinsic input and a specific recording location in the brain. (**E**) Effect size *R* for the recurrent connections captured by auto-regressive filters **A**.

Filter matrices **A** and **B** are unknown and can be estimated from the observed history of **x***(t)* and **y***(t)* using ordinary least squares (OLS). The objective for the optimal model is to minimize the power of the unobserved innovation process **e***(t)*, that is the summed squares:

$$\sigma^2 = \frac{1}{T} \sum_{t=1}^{T} e(t)^2$$

## Granger analysis

The innovation is also the prediction error, for predicting *y(t)* from the past *y(t-1)* and input *x(t)*. In the Granger formalism, the prediction error is calculated with all predictors included (error of the full model, $\sigma_f$) or with individual dimensions in *y(t-1)* or *x*(t) omitted from the prediction (error of the reduced models, $\sigma_r$) (*Granger, 1969*). To quantify the 'effect' of the specific dimension, one can take the ratio of these errors (*Geweke, 1982*), leading to the test statistic *D* known as the 'deviance'. When the number of samples *T* is large, the deviance follows the Chi-square distribution with cumulative density *F*, from which one can compute a p-value:

$$D = T \log(\sigma_r^2/\sigma_f^2)$$
$$p = 1 - F(D, T)$$
$$R^2 = 1 - e^{-D/T}$$

The p-value quantifies the probability that a specific connection in either **A** or **B** is zero. Therefore, *D*, *p* and *R²* all have dimensions $[d_y, d_y]$ or $[d_y, d_x]$ for **A** or **B** respectively. The 'generalized' *R²* (*Magee, 1990*) serves as a measure of effect size, capturing the strength of each connection. While this Granger formalism is well established in the context of estimating **A**, that is VAR models, to our knowledge, have not been used in the context of estimating **B**, that is VARX or TRF models.

## Overall system response

The overall brain response to the stimulus for the VARX model is given by the system impulse response (written here in the *z*-domain, or Fourier domain):

$$\mathbf{H} = (1 - \mathbf{A})^{-1}\mathbf{B}$$

What we see here is that the system response **H** is factorized into an autoregressive (AR) filter **A** and a moving average (MA) filter **B**. When modeled as a single MA filter, the total system response has been called the 'multivariate Temporal Response Function' (mTRF) in the neuroscience community (*Crosse et al., 2016*). We found that the VARX estimate **H** is nearly identical to the estimated mTRF (*Parra et al., 2025*). In other words, **B** and **A** are a valid factorization of the mTRF into feed-forward extrinsic versus recurrent intrinsic effects.

Note that the extrinsic effects captured with filters **B** are specific (every stimulus dimension has a specific effect on each brain area), whereas the intrinsic dynamic propagates this initial effect to all connected brain areas via matrix **A**, effectively mixing and adding the responses of all stimulus dimensions. Therefore, this factorization separates stimulus-specific effects from the shared intrinsic dynamic.

## Relation to common neural signal models

The VARX model fits naturally into the existing family of models used for neural signals analysis. While they differ in the formulation and statistical assumptions, their defining equations have a similar general form with the attributes summarized in *Table 1*.

An important simplifying assumption for the mTRF, VAR, and VARX models is that *y*(t) is observable with additive normally distributed innovation. As a result, parameter estimation can use ordinary least squares, which is fast to compute. In contrast, GLM, DCM, and some variants of VAR models assume that *y*(t) is not directly observable and needs to be estimated in addition to the unknown parameters **A** or **B**. The same is true for the basic 'output error' model in linear systems theory (*Ljung, 1999*). This requires slower iterative algorithms, such as expectation maximization. As a result, these models are often limited to small networks of a few nodes to test specific alternative hypotheses (*Penny et al., 2004*) (The original DCM proposed for fMRI included an added complication of modeling the hemodynamic response, which amounts to adding a temporal filter to each output node and prior to adding observation noise). In contrast, here we will analyze up to 300 channels per patient to draw general conclusions about overall brain organization.

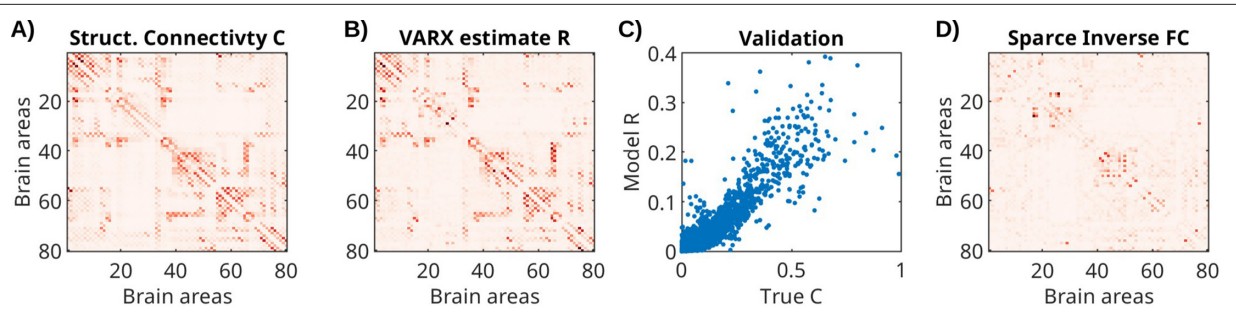

**Figure 7.** Structural connectivity of stimulated neural mass model for the whole brain, and estimated recurrent connectivity in VARX model. (**A**) True structural connectivity C used to simulate neural activity using a neural mass model with the neurolib python toolbox. Structural connectivity is based on diffusion tensor imaging data between 80 brain areas (called Cmat in neurolib). Here showing the square root of the 'Cmat' matrix for better visibility of small connectivity values. (**B**) Effect size estimate *R* for the recurrent connectivity matrix **A** of the VARX model on the simulated data. The diagonal in *R* is omitted as it is also missing in the structural connectivity Cmat. (**C**) Comparison of true and VARX estimate of connectivity. (**D**) Absolute value of the sparse-inverse functional connectivity (estimated using graphical lasso *Friedman et al., 2008*).

The online version of this article includes the following figure supplement(s) for figure 7:

**Figure supplement 1.** Connectivity of stimulated neural mass model.

**Figure supplement 2.** Connectivity of stimulated neural mass model with asymmetric structural connectivity.

## Validation of recurrent connectivity estimate with whole-brain neural mass model

To test the descriptive validity (*Bassett et al., 2018*) of the VARX model, we follow the approach of recovering structural connectivity from functional activity in simulation (*Honey et al., 2009*). Specifically, we will compare the recurrent connectivity **A** derived from brain activity simulated assuming a given structural connectivity, that is we ask, can the VARX model recover the underlying structural connectivity, at least in a simulated whole-brain model with known connectivity? We simulated neural activity for a whole-brain neural mass model (*Cakan et al., 2023*). We used the default model simulation of the neurolib python library (using their sample code for the 'ALNModel'), which is a mean-field approximation of adaptive exponential integrate-and-fire neurons. This model can generate simulated mean firing rates in 80 brain areas based on connectivity and delay matrices determined with DTI. We used 5 min of 'resting state' activity (no added stimulus, simulated at 0.1 ms resolution, subsequently downsampled to 100 Hz). The VARX model was estimated with $n_a$=2, and no input. The resulting estimate for **A** is dominated by the diagonal elements that capture the autocorrelation within brain areas (*Figure 7—figure supplement 1*). The true connectivity matrix from DTI (*Figure 7A*) is similar to the effect size estimate for the recurrent connections (*Figure 7B*). Following *Honey et al., 2009* we compare the two as a scatter plot (*Figure 7C*) and observe a Spearman correlation of 0.69. For comparison, we also used the sparse-inverse covariance method to recover connectivity from the correlation matrix (functional connectivity). This method is considered state-of-the-art as it is more sensitive than other methods in detecting structural connections (*Smith et al., 2011*) and uses the graphical lasso algorithm (*Chen, 2023*). The resulting connectivity estimate (*Figure 7D*) only achieves a Spearman correlation of 0.52. We note that the structural connectivity determined with DTI is largely symmetric. When enhancing the asymmetry, the VARX model is not as accurate, but correctly recovers the direction of the asymmetry (*Figure 7—figure supplement 2*).

## Intracranial EEG recordings and stimulus features

We analyzed intracranial EEG and simultaneous eye-tracking data recorded from patients (*N* = 26 recordings, mean age 38.69 years, age range 19–59 years, 11 females, *Appendix 2—table 1*) during rest and while they watched various video clips. Four out of 22 individual patients underwent two implantations and recordings at different times, resulting in a total of 26 recording sessions with a total of 5093 recording channels. The video clips included animations with speech ('Despicable Me', two different clips, 10 min each, in English and Hungarian), an animated short film with a mostly visual narrative and music, shown twice ('The Present', 4.3 min), and three clips of documentaries of macaques ('Monkey', 5 min each, without sound) (*Nentwich et al., 2023*). In addition to the clips from

the previous analysis, we included a movie clip of abstract animations ('Inscapes', 10 min) (*Vanderwal et al., 2015*), and an eyes-open resting state with maintained fixation ('Resting state', 5 min), and eyes-closed resting state ('Eyes Closed Rest', 5 min). In total, we recorded up to 64.7 min of data for each patient (*Appendix 2—table 1*).

Neural signals were preprocessed as previously described to reduce noise (*Nentwich et al., 2023*). We re-reference signals in a bipolar montage to ensure analysis of local activity. We analyze LFPs and BHA power. BHA is the power of the signal bandpass filtered between 70 and 150 Hz. We perform analysis on both signals after downsampling to 60 Hz. Example traces of $y(t)$ for LFP and BHA are shown in *Figure 6C*.

We extract six features of the movies that serve as external inputs for the VARX model: fixation onset, fixation novelty, film cuts, motion, sound envelope, and acoustic edges (*Figure 6B*). Fixation onset, fixation novelty, film cuts, and acoustic edges are represented in $x(t)$ as pulse trains with pulses occurring at the time of these events (*Nentwich et al., 2023*). Motion and the sound envelope are continuous regressors. Motion is the average optic flow across each frame (*Nentwich et al., 2023*). Fixation novelty is computed as the Euclidean distance between features of a convolutional neural network computed on pre- and post-fixation image patches. Fixation novelty aims to capture the change of the semantics of visual input across eye movements (*Nentwich et al., 2023*). The fixation novelty impulses are the same as the fixation regressor, but with their amplitude scaled by novelty. Sound envelope is computed as the absolute value of the Hilbert transform of the sound from the movie files. The envelope is downsampled to 60 Hz. Acoustic edges are peaks in the derivative of the sound envelope, representing rapid changes in the input (*Forseth et al., 2020*; *Oganian and Chang, 2019*). All videos and resting states include fixations. Since the visual environment is constant during fixations, there is no fixation novelty regressor. The video 'Inscapes', resting state, and eyes-closed rest do not include film cuts as external input. The 'Monkey' video clips, resting state, and eyes-closed rest do not include the sound envelope or acoustic edges as input features. Eyes-closed rest does not include any external inputs. When a feature is not available, it is replaced with features from a different recording. Therefore, the statistics of the feature are consistent, but not aligned to the neural recording. When comparing models with different features, we always keep the number of input variables consistent between models to avoid a bias by the number of free parameters of the model. Features that are not considered in the analysis are shuffled in time by a circular shift by half the duration of the signals.

The models were fitted to data with the MATLAB version of the publicly available VARX code (*Parra et al., 2025*) using conventional L2-norm regularization. The corresponding regularization parameter was set to $\lambda = 0.3$. For all analyses, we use filters of 600ms length for inputs ($n_b = 36$ samples for VARX models, $L = 36$ samples for mTRF models). Delays for connections between channels are set to 100 ms ($n_a = 6$ samples) for both LFP and BHA signals. Increasing the number of delays $n_a$, increases estimated effect size $R$ (*Appendix 3—figure 1A, B*); however, larger values lead to fewer significant connections (*Appendix 3—figure 1C*). Significance (p-value) is computed analytically, that is non-parametrically, based on deviance. Values around $n_a = 6$ time delays appear to be the largest model order supported by this statistical analysis.

Connectivity plots in *Figure 2* were created with routines from the nilearn toolbox (*Chamma et al., 2024*). We plot only significant connections (p < 0.001). Surface plots of T1w/T2w ratios and directionality of connections are created using the field-echos repository (*Gao et al., 2020a*; *Gao et al., 2020b*). T1w/T2w maps (*Glasser et al., 2016*) are obtained from the neuromaps repository (*Markello et al., 2024*; *Markello et al., 2022*), and transformed to the FreeSurfer surface using code from the neuromaps toolbox (*Robinson et al., 2018*; *Robinson et al., 2014*).

The length of responses for each channel in **B** and **H** to external inputs in *Figure 3* is computed with Matlab's findpeaks() function. This function returns the full width at half of the peak maximum minus baseline. Power in each channel is computed as the squares of the responses averaged over the time window that was analyzed (0–0.6 s).

To compare recurrent connectivity between movies and the resting state (in *Figure 2*), we compute VARX models in four different movie segments of 5 min length to match the length of the resting state recording. We use the first and second half of 'Despicable Me English', the first half of 'Inscapes', and one of the 'Monkey' movies. Eighteen patients include each of these recordings. For each recording in each patient, we compute the fraction of significant channels (p < 0.001) and average the effect

size across all channel pairs, excluding the diagonal. We test the difference between movies and resting state with linear mixed-effect models with stimulus as fixed effect (movie vs. rest) and patient as random effect (to account for the repeated measures for the different video segments), using MATLAB's fitlme() routine. For the analysis of asymmetry of recurrent connectivity (in *Figure 5*), we also used a mixed-effect model with T1w/T2w ratio as fixed effect and patients as random effect (to account for the repeated measures in multiple brain locations).

## Acknowledgements

We would like to thank Chris Honey for advice on the model validation with simulations and related references. We like to thank Behtash Babadi for help on the development of the Granger formalism for the VARX model. This work was supported in part by the NIH through grants P50 MH109429 and R01DC019979.

## Additional information

### Funding

| Funder | Grant reference number | Author |
|---|---|---|
| National Institutes of Health | P50MH109429 | Maximilian Nentwich<br>Marcin Leszczynski<br>Charles E Schroeder<br>Stephan Bickel<br>Lucas C Parra |
| National Institutes of Health | R01DC019979 | Maximilian Nentwich<br>Stephan Bickel |

The funders had no role in study design, data collection, and interpretation, or the decision to submit the work for publication.

### Author contributions

Maximilian Nentwich, Conceptualization, Data curation, Software, Formal analysis, Validation, Investigation, Visualization, Writing – original draft, Writing – review and editing; Marcin Leszczynski, Data curation, Validation, Investigation, Writing – review and editing; Charles E Schroeder, Supervision, Funding acquisition, Writing – review and editing; Stephan Bickel, Resources, Supervision, Funding acquisition, Project administration, Writing – review and editing; Lucas C Parra, Conceptualization, Software, Formal analysis, Supervision, Funding acquisition, Validation, Investigation, Visualization, Methodology, Writing – original draft, Project administration, Writing – review and editing

### Author ORCIDs

Maximilian Nentwich ⓘ https://orcid.org/0000-0002-9306-7591
Marcin Leszczynski ⓘ https://orcid.org/0000-0003-3172-4661
Lucas C Parra ⓘ https://orcid.org/0000-0003-4667-816X

### Ethics

This study was conducted in accordance with the Institutional Review Board at the Feinstein Institutes for Medical Research (Northwell Health), and informed consent was obtained prior to research testing.

Reviewer #1 (Public review): https://doi.org/10.7554/eLife.104996.3.sa1
Reviewer #2 (Public review): https://doi.org/10.7554/eLife.104996.3.sa2
Author response https://doi.org/10.7554/eLife.104996.3.sa3

## Additional files

### Supplementary files

MDAR checklist

## Data availability

The raw data reported in this study cannot be deposited in a public repository because of patient privacy concerns. To request access, contact The Feinstein Institutes for Medical Research, through Dr. Stephan Bickel. In addition, processed datasets derived from these data have been deposited at https://doi.org/10.17605/OSF.IO/VC25T. All original code has been deposited at https://github.com/MaxNentwich/varx_demo (copy archived at *MaxNentwich, 2025* and https://doi.org/10.5281/zenodo.15127333).

The following dataset was generated:

| Author(s) | Year | Dataset title | Dataset URL | Database and Identifier |
|-----------|------|---------------|-------------|--------------------------|
| Bickel S, Nentwich M | 2024 | VARX Granger Analysis | https://doi.org/10.17605/OSF.IO/VC25T | Open Science Framework, 10.17605/OSF.IO/VC25T |

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

# Appendix 1

## Intrinsic 'noise' and gain adaptation model

We see a clear drop in power of the innovation $e(t)$ over the entire spectrum analyzed for many channels (example of one patient in *Appendix 1—figure 1B*). When looking at the median electrode, we see that the effect is significant across all patients (*Appendix 1—figure 1C*, Wilcoxon signed rank test p = 0.005, *N* = 25 patients). The drop in variability is evident also in the raw signals $y(t)$ (*Appendix 1—figure 1C*, Wilcoxon signed rank test p = 0.011, *N* = 25). It is most pronounced in oscillatory bands. For instance, in the particular patient shown in *Appendix 1—figure 1A*, there is a clear reduction in theta band activity around 8 Hz. The specific bands differ across patients and channels (not shown). In total, even after oscillatory activity is modeled with the recursion filters *A*, there is a broadband reduction in power, while the relative noise power is unchanged (p > 0.1).

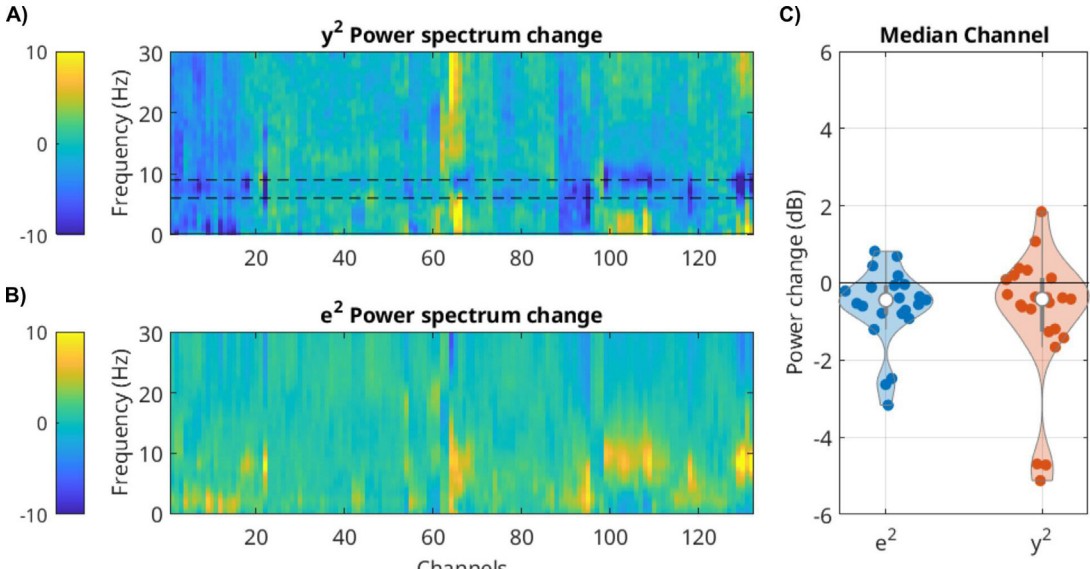

**Appendix 1—figure 1.** For local field potential (LFP), the power of the signal and innovation process drops during movies as compared to rest. Difference in LFP power between movies minus rest. Negative values indicate stronger power during rest. (**A**) Difference in power spectrum for the raw signal $y(t)$ for one patient. False color indicates change in the power spectrum in dB (blue hues indicate stronger power during rest). Dashed black lines bracket 5 and 11 Hz. (**B**) Difference in power spectrum for the innovation process $e(t)$. (**C**) Power difference movie minus rest for each of 25 patients (each point is the median over channels).

## Gain adaptation model

This is a VARX model, where at every step the activity is adapted to have constant power over a given time horizon. We implemented this as a divisive normalization with a running estimate of the power in the signal as follows:

$$\tilde{y}(t) = A * y(t-1) + B * x(t) + e(t)$$

$$\left|g(t)\right|^2 = (1-\gamma)\left|g(t-1)\right|^2 + \gamma\left|\tilde{y}(t)\right|^2$$

$$y(t) = \tilde{y}(t)/g(t)$$

The division with the gain $g(t)$ is element-wise. In the simulation here and in the main text, we used γ=0.001. This corresponds to power averaged over time with an exponential decay window with a time constant of $\tau=\Delta t\backslash\gamma$, where $\delta t$ is the sampling interval. The simulation of *Appendix 1—figure 2* shows that a signal generated with this gain adaptation mechanism will exhibit the reduction of relative power of innovation (noise quenching) when the stimulus comes on, relative to when there is no external stimulus. But this is only true if the underlying signal generation implements gain adaptation (compare *Appendix 1—figure 2C and D*).

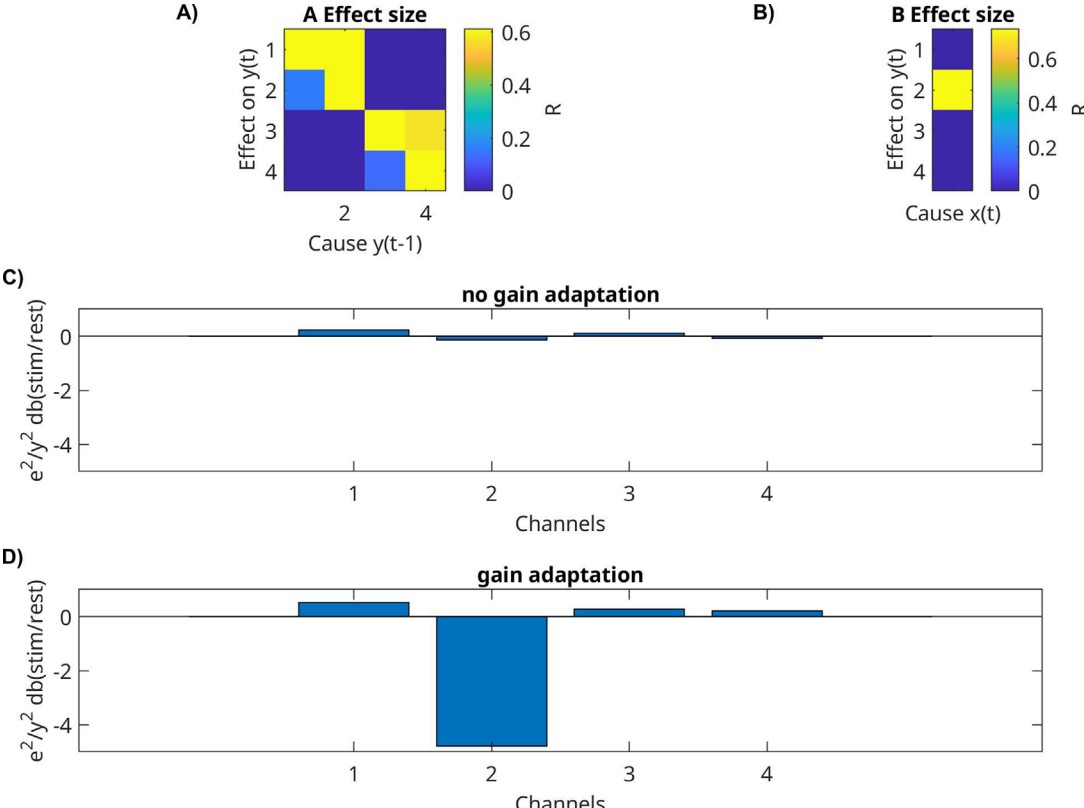

**Appendix 1—figure 2.** Gain adaptation on a toy example. In this small recurrent network, there are four nodes, with each of two nodes connected. Input only arrives at one node. Data are simulated with and without gain adaptation and then estimated with the VARX model. (**A**) Estimated recurrent connectivity. (**B**) Estimated input connectivity estimated during 'stimulus' condition. Relative power of the innovation **e**(t) (relative to signal **y**(t)) subtracting dB between stimulus − rest condition, (**C**) without and (**D**) with gain adaptation. 'Rest' here means that the input **x**(t) was zero.

# Appendix 2

## Demographics

**Appendix 2—table 1.** Demographics, length of recordings, and number of recording channels. Data from patients 9, 11, 18, and 23 were recorded from two reimplants each at different times.

| Patient ID | Age | Sex | Total length of recordings [min] | Number of channels |
|---|---|---|---|---|
| Pat_1 | 58 | M | 58.6 | 153 |
| Pat_2 | 22 | M | 48.6 | 164 |
| Pat_5 | 48 | M | 58.6 | 334 |
| Pat_6 | 36 | F | 58.6 | 189 |
| Pat_7 | 43 | M | 58.6 | 132 |
| Pat_8 | 41 | F | 64.7 | 154 |
| Pat_9 | 50 | M | 58.6 | 267 |
| Pat_9_02 | 51 | M | 58.6 | 271 |
| Pat_10 | 24 | M | 53.6 | 192 |
| Pat_11 | 37 | M | 58.6 | 192 |
| Pat_11_02 | 37 | M | 58.6 | 111 |
| Pat_12 | 52 | F | 48.6 | 198 |
| Pat_13_02 | 24 | M | 58.6 | 296 |
| Pat_14 | 20 | M | 58.6 | 207 |
| Pat_15 | 56 | M | 53.6 | 100 |
| Pat_16 | 43 | F | 39.3 | 261 |
| Pat_17 | 27 | F | 58.6 | 220 |
| Pat_18 | 28 | F | 58.6 | 227 |
| Pat_18_02 | 30 | F | 58.6 | 71 |
| Pat_19 | 46 | M | 48.6 | 230 |
| Pat_20 | 35 | F | 54.3 | 231 |
| Pat_21 | 48 | M | 58.6 | 323 |
| Pat_22 | 19 | M | 48.6 | 75 |
| Pat_23 | 36 | F | 54.4 | 175 |
| Pat_23_03 | 36 | F | 48.7 | 260 |
| Pat_24 | 59 | F | 39.3 | 60 |

# Appendix 3

## Determination of the number of delays

On the data for a single patient (Pat_1), we determine the effects of different numbers of delays $n_a$ for intrinsic connectivity. Increasing the number of delays increases $R$ values overall, but decreases the number of significant connections (*Appendix 3—figure 1*).

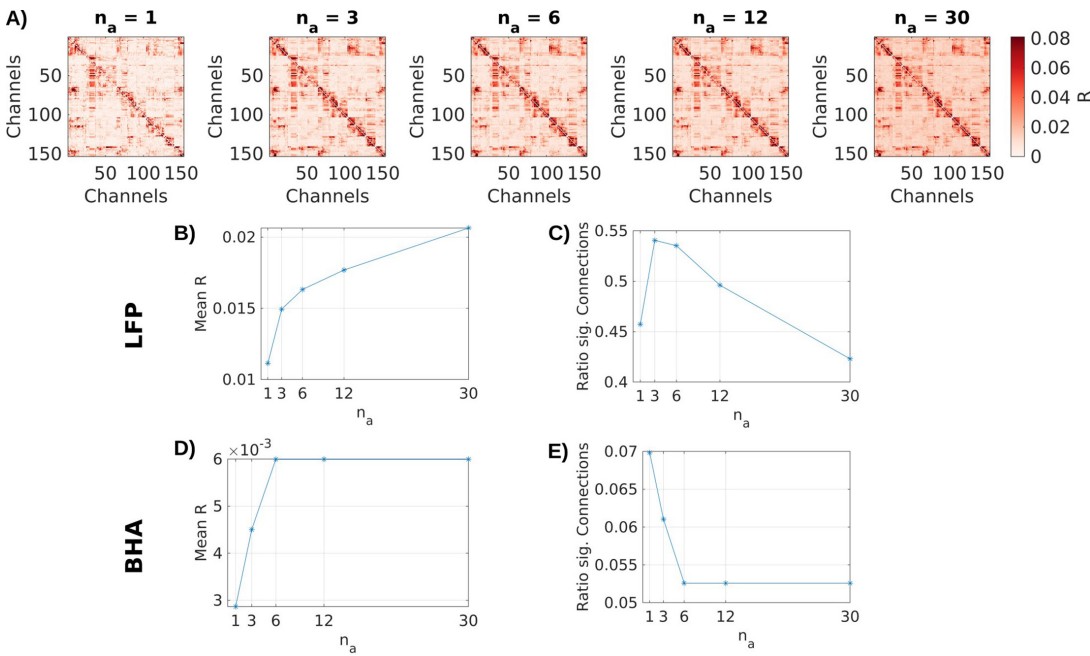

**Appendix 3—figure 1.** Choice of $n_a$. (**A**) Effect size $R$ of connections in local field potential (LFP) data for an example patient using different values for the delays $n_a$. (**B**) Mean effect size $R$ and (**C**) ratio of significant ($p < 0.001$) channels across all channels for different $n_a$ for LFP data. (**D**) and (**E**) in broadband high-frequency activity (BHA).

