## [Editor Report · eLife Assessment]

This manuscript presents an interesting new framework (VARX) for simultaneously quantifying effective connectivity in brain activity during sensory stimulation and how that brain activity is being driven by that sensory stimulation. The reviewers thought the model was original and its conclusion that intrinsic connectivity is reduced (rather than increased) during sensory stimulation is very interesting, but that for ideal performance, one must specify all sensory features in the model, which is not possible. Overall, however, this work is **important** with **convincing** evidence for its conclusions - it will be of interest to neuroscientists working on brain connectivity and dynamics.

---

## [Referee Report · Reviewer #1 (Public review)]

This manuscript presents an interesting new framework (VARX) for simultaneously quantifying effective connectivity in brain activity during sensory stimulation and how that brain activity is being driven by that sensory stimulation. The core idea is to combine the Vector Autoregressive model that is often used to infer Granger-causal connectivity in brain data with an encoding model that maps the features of a sensory stimulus to that brain data. The authors do a nice job of explaining the framework. And then they demonstrate its utility through some simulations and some analysis of real intracranial EEG data recorded from subjects as they watched movies. They infer from their analyses that the functional connectivity in these brain recordings is essentially unaltered during movie watching, that accounting for the driving movie stimulus can protect one against misidentifying brain responses to the stimulus as functional connectivity, and that recurrent brain activity enhances and prolongs the putative neural responses to a stimulus.

This manuscript presents an interesting new framework (VARX) for simultaneously quantifying effective connectivity in brain activity during sensory stimulation and how that brain activity is being driven by that sensory stimulation. Overall, I thought this was an interesting manuscript with some rich and intriguing ideas.

Comments on revisions:'

The responses to the previous comments are very helpful. I think the manuscript does a nice job now of presenting its interesting findings in a convincing and measured manner.

I had only one small remaining suggestion - to maybe link the finding of reduced intrinsic connectivity during stimulation to previous work on that topic. I thought of Nauhaus et al., Nature Neurosci, 2009.

---

## [Referee Report · Reviewer #2 (Public review)]

Summary:

The authors apply the recently developed VARX model, which explicitly models intrinsic dynamics and the effect of extrinsic inputs, to simulated data and intracranial EEG recordings. This method provides a directed method of 'intrinsic connectivity'. They argue this model is better suited to the analysis of task neuroimaging data because it separates the intrinsic and extrinsic activity. They show: that intrinsic connectivity is largely unaltered during a movie-watching task compared to eyes open rest; intrinsic noise is reduced in the task; and there is intrinsic directed connectivity from sensory to higher-order brain areas.

Strengths:

(1) The paper tackles an important issue with an appropriate method.

(2) The authors validated their method on data simulated with a neural mass model.

(3) They use intracranial EEG, which provides a direct measure of neuronal activity.

(4) Code is made publicly available and the paper is written well.

Comments on revisions:'

The authors have addressed my comments.

---

## [Author Response]

The following is the authors’ response to the original reviews

**Public Reviews:**

**Reviewer #1 (Public review):**
This manuscript presents an interesting new framework (VARX) for simultaneously quantifying effective connectivity in brain activity during sensory stimulation and how that brain activity is being driven by that sensory stimulation. The core idea is to combine the Vector Autoregressive model that is often used to infer Granger-causal connectivity in brain data with an encoding model that maps the features of a sensory stimulus to that brain data. The authors do a nice job of explaining the framework. And then they demonstrate its utility through some simulations and some analysis of real intracranial EEG data recorded from subjects as they watched movies. They infer from their analyses that the functional connectivity in these brain recordings is essentially unaltered during movie watching, that accounting for the driving movie stimulus can protect one against misidentifying brain responses to the stimulus as functional connectivity, and that recurrent brain activity enhances and prolongs the putative neural responses to a stimulus.This manuscript presents an interesting new framework (VARX) for simultaneously quantifying effective connectivity in brain activity during sensory stimulation and how that brain activity is being driven by that sensory stimulation. Overall, I thought this was an interesting manuscript with some rich and intriguing ideas. That said, I had some concerns also - one potentially major - with the inferences drawn by the authors on the analyses that they carried out.Main comments:(1) My primary concern with the way the manuscript is written right now relates to the inferences that can be drawn from the framework. In particular, the authors want to assert that, by incorporating an encoding model into their framework, they can do a better job of accounting for correlated stimulus-driven activity in different brain regions, allowing them to get a clearer view of the underlying innate functional connectivity of the brain. Indeed, the authors say that they want to ask "whether, after removing stimulus-induced correlations, the intrinsic dynamic itself is preserved". This seems a very attractive idea indeed. However, it seems to hinge critically on the idea of fitting an encoding model that fully explains all of the stimulus-driven activity. In other words, if one fits an encoding model that only explains some of the stimulus-driven response, then the rest of the stimulus-driven response still remains in the data and will be correlated across brain regions and will appear as functional connectivity in the ongoing brain dynamics - according to this framework. This residual activity would thus be misinterpreted. In the present work, the authors parameterize their stimulus using fixation onsets, film cuts, and the audio envelope. All of these features seem reasonable and valid. However, they surely do not come close to capturing the full richness of the stimuli, and, as such, there is surely a substantial amount of stimulus-driven brain activity that is not being accounted for by their "B" model and that is being absorbed into their "A" model and misinterpreted as intrinsic connectivity. This seems to me to be a major limitation of the framework. Indeed, the authors flag this concern themselves by (briefly) raising the issue in the first paragraph of their caveats section. But I think it warrants much more attention and discussion.

We agree. One can never be sure that all stimulus induced correlation is accounted for. We now formulate our question more cautiously:

“We will ask here whether, after removing some of the stimulus-induced correlations, the intrinsic dynamic is similar between stimulus and rest conditions.”

We also highlight that one may expect the opposite result of what we found:

“A general observation of these studies is that a portion of the functional connectivity is preserved between rest and stimulus conditions, while some aspects are altered by the perceptual task [12,16], sometimes showing increased connectivity during the stimulus.[15].”

We have added a number of additional features (acoustic edges, fixation novelty, and motion) and more carefully characterize how much “connectivity” each one explains in the neural data:

“Removing any of the input features increased the effect size of recurrent connections compared to a model with all features (Fig. S4). We then cumulatively added each feature to the VARX model. Effect size monotonically decreases with each feature added (Fig. 3F). Decreases of effect size are significant when adding film cuts (ΔR=-3.6*10^-6^, p<0.0001, N=26, FDR correction, α=0.05) and the sound envelope (ΔR=-3.59*10^-6^, p=0.002, N=26, FDR correction, α=0.05). Thus, adding more input features progressively reduces the strength of recurrent “connections”.”

We also added more data to the analysis comparing movies vs rest. We now use 4 different movie segments instead of 1 and find reduced recurrent connectivity during movies:

“The number of significant recurrent connections in were significantly reduced during movie watching compared to rest (Fig. 4C, fixed effect of stimulus: beta = -3.8*10^-3^, t(17) = -3.9, p<0.001), as is the effect size *R* (Fig. 4D, fixed effect of stimulus: beta = -2.5*10^-4^, t(17) = -4.1, p<0.001).”

The additional analysis is described in the Methods section:

“To compare recurrent connectivity between movies and the resting-state, we compute VARX models in four different movie segments of 5 minutes length to match the length of the resting state recording. We use the first and second half of ‘Despicable Me English’, the first half of ‘Inscapes’ and one of the ‘Monkey’ movies. 18 patients include each of these recordings. For each recording in each patient we compute the fraction of significant channels (p<0.001) and average the effect size *R* across all channel pairs, excluding the diagonal. We test the difference between movies and resting-state with linear mixed-effect models with stimulus as fixed effect (movie vs rest), and patient as random effect, using matlab’s fitlme() routine.”

We had already seen this trend of decreasing connectivity during movie watching before, and reported on it cautiously as “largely unaltered”. We updated the Abstract correspondingly from “largely unaltered” to “reduced”:

“We also find that the recurrent connectivity during rest is reduced during movie watching.”

We mentioned this possibility in the Discussion before, namely, that additional input features may reduce recurrent connectivity in the model, and therefore show a difference. We discuss this result now as follows:

“The stimulus features we included in our model capture mostly low-level visual and auditory input. It is possible that regressing out a richer stimulus characterization would have removed additional stimulus-induced correlation. While we do not expect that this would change the overall effect of a reduced number of “connections” during movie watching compared to resting state, the interpretation of changes in specific connections will be affected by the choice of features. For example, in sensory cortices, higher recurrent connectivity in the LFP during rest would be consistent with the more synchronized state we saw in rest, as reflected by larger oscillatory activity. Synchronization in higher-order cortices, however, is expected to be more strongly influenced by semantic content of external input.”

In the Discussion we expand on what might happen if additional stimulus features were to be included into the model:

“Previous literature does often not distinguish between intrinsic dynamics and extrinsic effects. By factoring out some of the linear effects of the external input we conclude here that recurrent connectivity is reduced in average. From our prior work49, we know that the stimulus features we included here capture a substantial amount of variance across the brain in intracranial EEG. Arguably, however, the video stimuli had rich semantic information that was not captured by the low-level features used here. Adding such semantic features could have further reduced shared variance, and consequently further reduced average recurrent connectivity in the model.”

“Similarities and differences between rest and movie watching conditions reported previously, do not draw a firm conclusion as to whether overall “functional connectivity” is increased or reduced. Results seem to depend on the time scale of neural activity analyzed, and the specific brain networks [12,16,63]. However, in fMRI, the conclusion seems to be that functional connectivity during movies is stronger than during rest[15], which likely results from stimulus induced correlations. The VARX model can remove some of the effects of these stimuli, revealing that average recurrent connectivity may be reduced rather than increased during stimulus processing.”

And in the conclusion we now write:

“The model revealed a small but significant decrease of recurrent connectivity when watching movies.”

(2) Related to the previous comment, the authors make what seems to me to be a complex and important point on page 6 (of the pdf). Specifically, they say "Note that the extrinsic effects captured with filters B are specific (every stimulus dimension has a specific effect on each brain area), whereas the endogenous dynamic propagates this initial effect to all connected brain areas via matrix A, effectively mixing and adding the responses of all stimulus dimensions. Therefore, this factorization separates stimulus-specific effects from the shared endogenous dynamic." It seems to me that the interpretation of the filter B (which is analogous to the "TRF") for the envelope, say, will be affected by the fact that the matrix A is likely going to be influenced by all sorts of other stimulus features that are not included in the model. In other words, residual stimulus-driven correlations that are captured in A might also distort what is going on in B, perhaps. So, again, I worry about interpreting the framework unless one can guarantee a near-perfect encoding model that can fully account for the stimulus-driven activity. I'd love to hear the authors' thoughts on this. (On this issue - the word "dominates" on page 12 seems very strong.)

This is an interesting point we had not thought about. After some theoretical considerations and some empirical testing we conclude that the effect of missing inputs is relevant, but can be easily anticipated.

We have added the following to the Results section explaining and demonstrated empirically the effects of adding features and signals to the model:

“As with conventional linear regression, the estimate in **B** for a particular input and output channel is not affected by which other signals are included in \begin{document}$\mathbf{x}(t)$\end{document} or \begin{document}$\mathbf{y}(t)$\end{document}, provided those other inputs are uncorrelated. We confirmed this here empirically by removing dimensions from \begin{document}$\mathbf{y}(t)$\end{document} (Fig. S11A), and by adding uncorrelated input to **B**, we do not require all possible stimulus features and all brain activity to be measured and included in the model. In contrast, **B** does vary when correlated inputs are added to \begin{document}$\mathbf{x}(t)$\end{document} (Fig. S11B, adding fixation onset does not affect the estimate for auditory envelope responses). In other words, to estimate \begin{document}$\mathbf{x}(t)$\end{document} (Fig. S11C, adding acoustic edges changes the auditory envelope response). Evidently the auditory envelope and acoustic edges are tightly coupled in time, whereas fixation onset is not. When a correlated input is missing (acoustic edges) then the other input (auditory envelope) absorbs the correlated variance, thus capturing the combined response of both.”

(3) Regarding the interpretation of the analysis of connectivity between movies and rest... that concludes that the intrinsic connectivity pattern doesn't really differ. This is interesting. But it seems worth flagging that this analysis doesn't really account for the specific dynamics in the network that could differ quite substantially between movie watching and rest, right? At the moment, it is all correlational. But the dynamics within the network could be very different between stimulation and rest I would have thought.

As discussed above, with more data and additional stimulus features we now see detectable changes in the connectivity. The example in Figure 4G also shows that specific connections may change in different directions, while overall the strength of connections slightly decreases during movie watching compared to rest. We added the following to the results:

“While the effect size decreases on average, there is some variation across different brain areas (Fig. 4E-G).”

But even if the connectivity were unchanged, the activity on this network can be different with varying inputs. We actually also saw that there were changes in the variability of activity (Figs. 6 and S13) that may point to non-linear effects. It seems that injecting the input will cause an overall change in power, which can be explained by a relatively simple non-linear gain adaptation. These effects are already discussed at some length in the paper.

(4) I didn't really understand the point of comparing the VARX connectivity estimate with the spare-inverse covariance method (Figure 2D). What was the point of this? What is a reader supposed to appreciate from it about the validity or otherwise of the VARX approach?

We added the following motivation and clarification on this topic:

“To test the descriptive validity [43] of the VARX model we follow the approach of recovering structural connectivity from functional activity in simulation. [44] Specifically, we will compare the recurrent connectivity **A** derived from brain activity simulated assuming a given structural connectivity, i.e. we ask, can the VARX model recover the underlying structural connectivity, at least in a simulated whole-brian model with known connectivity? … For comparison, we also used the sparse-inverse covariance method to recover connectivity from the correlation matrix (functional connectivity). This method is considered state-of-the-art as it is more sensitive than other methods in detecting structural connections [48]”

(5) I think the VARX model section could have benefitted a bit from putting some dimensions on some of the variables. In particular, I struggled a little to appreciate the dimensionality of A. I am assuming it has to involve both time lags AND electrode channels so that you can infer Granger causality (by including time) between channels. Including a bit more detail on the dimensionality and shape of A might be helpful for others who want to implement the VARX model.

Your assumption is correct. We added the following to make this easier for readers:

“Therefore, **A** has dimensions \begin{document}$\left[d_{y}, d_{y}, n_{a}\right]$\end{document}
**B** has dimensions \begin{document}$\left[d_{y}, d_{x}, n_{b}\right]$\end{document}, where \begin{document}$d_{y}, d_{x}$\end{document} are the dimensions of \begin{document}$\mathbf{y}(t)$\end{document} and \begin{document}$\mathbf{x}(t)$\end{document} respectively.”

(6) A second issue I had with the inferences drawn by the authors was a difficulty in reconciling certain statements in the manuscript. For example, in the abstract, the authors write "We find that the recurrent connectivity during rest is largely unaltered during movie watching." And they also write that "Failing to account for ... exogenous inputs, leads to spurious connections in the intrinsic "connectivity".

Perhaps this segment of the abstract needed more explanation. To enhance clarity we have also changed the ordering of the findings. Hopefully this is more clear now:

“This model captures the extrinsic effect of the stimulus and separates that from the intrinsic effect of the recurrent brain dynamic. We find that the intrinsic dynamic enhances and prolongs the neural responses to scene cuts, eye movements, and sounds. Failing to account for these extrinsic inputs, leads to spurious recurrent connections that govern the intrinsic dynamic. We also find that the recurrent connectivity during rest is reduced during movie watching.”

**Reviewer #2 (Public review):**
Summary:The authors apply the recently developed VARX model, which explicitly models intrinsic dynamics and the effect of extrinsic inputs, to simulated data and intracranial EEG recordings. This method provides a directed method of 'intrinsic connectivity'. They argue this model is better suited to the analysis of task neuroimaging data because it separates the intrinsic and extrinsic activity. They show: that intrinsic connectivity is largely unaltered during a movie-watching task compared to eyes open rest; intrinsic noise is reduced in the task; and there is intrinsic directed connectivity from sensory to higher-order brain areas.Strengths:(1) The paper tackles an important issue with an appropriate method.(2) The authors validated their method on data simulated with a neural mass model.(3) They use intracranial EEG, which provides a direct measure of neuronal activity.(4) Code is made publicly available and the paper is written well.Weaknesses:It is unclear whether a linear model is adequate to describe brain data. To the author's credit, they discuss this in the manuscript. Also, the model presented still provides a useful and computationally efficient method for studying brain data - no model is 'the truth'.

We fully agree and have nothing much to add to this, except to highlight the benefit of a linear model even as explanation for non-linear phenomena:

“The [noise-quenching] effect we found here can be explained by a VARX model with the addition of a divisive gain adaptation mechanism … The noise-quenching result and its explanation via gain adaptation shows the benefit of using a parsimonious linear model, which can suggest nonlinear mechanisms as simple corrections from linearity.”

Appraisal of whether the authors achieve their aims:

As a methodological advancement highlighting a limitation of existing approaches and presenting a new model to overcome it, the authors achieve their aim. Generally, the claims/conclusions are supported by the results.

The wider neuroscience claims regarding the role of intrinsic dynamics and external inputs in affecting brain data could benefit from further replication with another independent dataset and in a variety of tasks - but I understand if the authors wanted to focus on the method rather than the neuroscientific claims in this manuscript.

We fully agree. We added the following to the Discussion section:

“Future studies should test if our findings replicate in an independent iEEG datasets, including active tasks and whether they generalize to other neuroimaging modalities.”

Impact:The authors propose a useful new approach that solves an important problem in the analysis of task neuroimaging data. I believe the work can have a significant impact on the field.
**Recommendations for the authors:**

**Reviewer #1 (Recommendations for the authors):**
Minor comments:(1) Did you mean "less" or "fewer" in the following sentence "..larger values lead to overfitting, i.e. less significant connections..."?

We mean fewer. Thanks for catching this.

(2) I didn't see any equations showing how the regularization parameter lambda is incorporated into the framework.

We prefer the math and details of the algorithm to an earlier paper that has now been published. Instead we added the following clarification:

“The VARX models were fitted to data with the matlab version of the code31 using conventional L2-norm regularization. The corresponding regularization parameter was set to 𝜆=0.3.”

(3) I think some readers of this might struggle to understand the paragraph beginning"Connectivity plots are created with nilearn's plot_connectome() function...". It's all quite opaque for the uninitiated.

Agreed. We now write more simply:

“Connectivity plots in Fig. 4 were created with routines from the nilearn toolbox [51].”

(4) The paragraph beginning "The length of responses for Figure 5..." is also very opaque and could do with being explained more fully. Or this text could be removed from the methods and incorporated into the relevant results section where you actually discuss this analysis.

Thank you for flagging this. We expand on the details in the Methods as follows:

“The length of responses for each channel in B and H to external inputs in Fig. 5 is computed with Matlab's findpeaks() function. This function returns the full-width at half of the peak maximum minus baseline. Power in each channel is computed as the squares of the responses averaged over the time window that was analyzed (0-0.6s).”

(5) I think adding some comments to the text or caption related to Figures 3C and 3D would be helpful so readers can understand these numbers a bit better. One seems to be the delta log p value and the other is the delta ratio. What does positive or negative mean? Readers might appreciate a little more help.

We expanded it as follows, hopefully this helps:

“(C) difference of log for VAX model without minus with inputs (panel A - B). Both models are fit to the same data. (D) Thresholding panels A and B at p<0.0001 gives a fraction of significant connections. Here we show the fraction of significant channels for models with and without input. Each line is a patient with color indicating increase or decrease (E) Mean over all channels for VARX models with and without inputs. Each line is a patient.”

(6) It is not clear what the colors mean in Figures 4 E, F, G.

We updated the color scheme for those figure panels and carefully explained it in the caption. Please see the manuscript for updated figure 4.

(7) It might be nice to slightly unpack what you mean by the "variability of the internal dynamic" and why it can be equated with the power of the innovation process.

In the methods we added the following clarification right after defining the VARX model:

“The innovation process \begin{document}$\mathbf{e}(t)$\end{document} captures the internal variability of the model. Without it, repeating the same input \begin{document}$\mathbf{x}(t)$\end{document} would always result in a fixed deterministic output \begin{document}$\mathbf{y}(t)$\end{document}.”

In the results section we added the following:

“As a metric of internal variability we measured the power of the intrinsic innovation process , which captures the unobserved “random” brain activity which leads to variations in the responses.”

(8) Typos etc.a) "... has been attributed to variability of ongoing dynamic"b) The manuscript refers to a Figure 3G, but there is no Figure 3G.c) n_a = n_a = 1. Is that a typo?d) fiction

Thank you for catching these. We fixed them.

**Reviewer #2 (Recommendations for the authors):**
(1) I'm curious about the authors' opinions on the conditions studied. Naively, eyes open rest and passive movie watching seem like similar conditions - were the authors expecting to see a difference with VARX? Do the authors expect that they would see bigger differences when there is a larger difference in sensory input, e.g. eyes closed rest vs movie watching? Given the authors are arguing the need to explicitly model external inputs, a real data example contrasting two very different external inputs might better demonstrate the model's utility.

Thank you for this suggestion. We added an analysis of eyes-closed rest recordings, available in 8 patients (Fig. S8). The difference between movie and rest is indeed more pronounced than for eyes open rest. The result is described in the methods:

“In a subset of patients with eyes-closed resting state we find the same effect, that is qualitatively more pronounced (Fig. S8).”

This complements our updated finding of a difference between movie and eyes-open rest that does show a significant difference after adding more data to this analysis. The results have been updated as following

“The number of significant recurrent connections in were significantly reduced during movie watching compared to rest Fig. 4C, fixed effect of stimulus:

beta = -3.8*10^-3^, t(17) = -3.9, p<0.001, as is the effect size *R* (Fig. 4D, fixed effect of stimulus: beta = -2.5*10^-4^, t(17) = -4.1, p<0.001).”

The abstract has been updated accordingly:

“We also find that the recurrent connectivity during rest is reduced during movie watching.”

(2) It would also have been interesting to see how the proposed model compares to DCM - however, I understand if the authors wanted to focus on their model rather than a comparison with other models.

We did not try the DCM for a number of reasons. (1) it does not allow for delays in the model dynamic (i.e. the entire time course of the response has to be captured by the recurrent dynamic of a single time step A). 2. It is computationally prohibitive and would not allow us to analyze large channel counts. 3. The available code is custom made for fMRI or EEG analysis with very specified signal generation models that do not obviously apply to iEEG. We added the following to the Discussion of the CDM:

“Similar to the VARX model, DCM includes intrinsic and extrinsic effects ***A*** and ***B***. However, the modeling is limited to first-order dynamics (i.e. *ηa*=*ηb*=1). Thus, prolonged responses have to be entirely captured with a first-order recurrent ***A***. … In contrast, here we have analyzed up to 300 channels per subject across the brain, which would be prohibitive with DCM. By analyzing a large number of recordings we were able to draw more general conclusions about whole-brain activity.”

(3) I believe improving the consistency of the terminology used would improve the manuscript:a) Intrinsic dynamics vs intrinsic connectivity vs recurrent connectivity:- The term 'intrinsic dynamic' is first introduced in paragraph 3 of the introduction. An explicit definition of is meant by this term would benefit the manuscript.- Sometimes the terminology changes to 'intrinsic connectivity' or 'recurrent connectivity'. An explicit definition of these terms (if they refer to different things) would also benefit the manuscript.

We had used the term “intrinsic” and “recurrent” interchangeably. We now try to mostly say “intrinsic dynamic” when we talk about the more general phenomenon or recurrent brain dynamic, while using “recurrent connectivity” when we refer to the model parameters A.

We provide now a definition already at the start of the Abstract:

“Sensory stimulation of the brain reverberates in its recurrent neural networks. However, current computational models of brain activity do not separate immediate sensory responses from this intrinsic dynamic. We apply a vector-autoregressive model with external input (VARX), combining the concepts of “functional connectivity” and “encoding models”, to intracranial recordings in humans. This model captures the extrinsic effect of the stimulus and separates that from the intrinsic effect of the recurrent brain dynamic.”

And at the start of the introduction:

“The primate brain is highly interconnected between and within brain areas. … We will refer to the dynamic driven by this recurrent architecture as the intrinsic dynamic of the brain.”

b) Intrinsic vs Endogenous and Extrinsic vs Exogenous:- Footnote 1 defines the 'intrinsic' and 'extrinsic' terminology.- However, there are instances where the authors switch back to endogenous/exogenous.- Methods section: "Overall system response", paragraph 2.- Results section: "Recurrent dynamic enhances and prolongs stimulus responses".- Conclusions section.

With a foot in both neuroscience and systems identification, it’s a hard habit to break. Thanks for catching it. We searched and replaced all instances of endogenous and exogenous.

(4) Methods:a) The model equation would be clearer if the convolution was written out fully. (I had to read reference 1 to understand the model.).

We now spell out the full equation and hope it's not too cumbersome to read:

“For the th signal channel the recurrence of the VARX model is given by:

\begin{document}$\mathbf{y}_{i}(t)=\sum_{j=1}^{d_{y}} \sum_{\tau=1}^{n_{a}} \mathbf{A}_{i j}(\tau) \mathbf{y}_{j}(t-\tau)+\sum_{j=1}^{d_{x}} \sum_{\tau=0}^{n_{b}} \mathbf{B}_{i j}(\tau) \mathbf{x}_{j}(t-\tau)+\mathbf{e}_{i}(t)$\end{document}”

b) How is an individual dimension omitted in the reduced model, are the values in the y, x set to zero?

No, it is actually removed from the linear prediction. We added:

“… omitted from the prediction …”

c) "The p-value quantifies the probability that a specific connection in A or B is zero" - for each of n_a/n_b filters?d) It should be clarified that D is a vector.

We hope the following clarification addresses both these questions:

“The p-value quantifies the probability that a specific connection in either ***A*** or ***B*** is zero. Therefore, *D,P* and *R2* all have dimensions \begin{document}$\left[d_{v}, d_{v}\right]$\end{document} or \begin{document}$d_{y}, d_{x}$\end{document} for ***A*** or ***B*** respectively.”

(5) Results:a) Stimulus-induced reduction of noise in the intrinsic activity: would be good to define the frequency range for theta and beta in paragraph 2.

Added.

b) Neural mass model simulation:- A brief description of what was simulated is needed.

We basically ran the sample code of the neurolib library. With that in mind maybe the description we already provide is sufficient:

“We used the default model simulation of the neurolib python library (using their sample code for the “ALNModel”), which is a mean-field approximation of adaptive exponential integrate-and-fire neurons. This model can generate simulated mean firing rates in 80 brain areas based on connectivity and delay matrices determined with diffusion tensor imaging (DTI). We used 5 min of “resting state” activity (no added stimulus, simulated at 0.1ms resolution, subsequently downsampled to 100Hz).”

- It's not clear to me why the A matrix should match the structural connectivity.

We added the following introduction to make the purpose of this simulation clear:

“To test the descriptive validity [43] of the VARX model we follow the approach of recovering structural connectivity from functional activity in simulation. [44] Specifically, we will compare the “connectivity” **A** derived from brain activity simulated assuming a given structural connectivity, i.e. we ask, can the VARX model recover the underlying structural connectivity, at least in a simulated whole-brian model with known connectivity?”

- It would be interesting to see the inferred A matrix.

We added a Supplement figure for this and the following:

“The VARX model was estimated with *na*=2, and no input. The resulting estimate for **A** is dominated by the diagonal elements that capture the autocorrelation within brain areas (Fig. S1).”

- How many filters were used here?

No input filters were used for this simulation:

We used 5 min of “resting state” activity (no added stimulus, simulated at 0.1ms resolution, subsequently downsampled to 100Hz).

c) Intracranial EEG:- It's not clear how overfitting was measured and how the selection of the number of filters (n_a and n_b) was done.

We have removed the statement about overfitting. Mostly the word is used in the context of testing on a separate dataset, which we did not do here. So this “overfitting” can be confusing. Instead we used the analytic p-value as indication that a larger model order is not supported by the data. We write this now as follows:

“Increasing the number of delays *na*, increases estimated effect size *R* (Fig. S3A,B), however, larger values lead to fewer significant connections (Fig. S3C). Significance (p-value) is computed analytically, i.e. non-parametrically, based on deviance. Values around *na*=6 time delays appear to be the largest model order supported by this statistical analysis.”

d) Figure 1:- Typo: "auto-regressive"

Fixed. Thanks for catching that.

- LFP and BHA in C are defined much later in the text, would be useful to define these in the caption. o Shouldn't B (the VARX model parameter) be a 2x3 matrix for different time lags?

Hopefully the following clarifications address both these points:

“(C) Example of neural signal y(t) recorded at a single location in the brain. We will analyze local field potentials (LFP) and broad-band high frequency activity (BHA) in separate analyses. (D) Examples of filters **B** for individual feed-forward connections between an extrinsic input and a specific recording location in the brain.”

(6) Discussion:I could not find Muller et al 2016 listed in the references.

Added. Thanks for catching that omission.

Additional edits prompted by reviewers, but not in the context of any particular comment.

While reviewers did not raise this following point, we felt the need clarify the terminology in the Methods to make sure there is not misunderstanding in the proposed interpretation of the model:

“We will refer to the filters in matrix **A** and **B** and as recurrent and feed-forward “connections”, but avoid the use of the word “causal” which can be misleading.”

In addressing questions to Figure 4, we noticed that there is quite a bit of variability across patients, so the analysis for Figure 4 and 7 which combines data across patients now accounts for a random effect of patient (previously we have used mean values for repeated measures). We added the following to the Methods to explain this:

“To compare recurrent connectivity between movies and the resting-state (in Fig. 4), we compute VARX models in four different movie segments of 5 minutes length to match the length of the resting state recording. We use the first and second half of ‘Despicable Me English’, the first half of ‘Inscapes’ and one of the ‘Monkey’ movies. 18 patients include each of these recordings. For each recording in each patient we compute the fraction of significant channels (p<0.001) and average the effect size *R* across all channel pairs, excluding the diagonal. We test the difference between movies and resting-state with linear mixed-effect models with stimulus as fixed effect (movie vs rest), and patient as random effect (to account for the repeated measures for the different video segments), using matlab’s fitlme() routine. For the analysis of asymmetry of recurrent connectivity (in Fig. 4) we also used a mixed-effect model with T1w/T2w ratio as fixed effect and patients as random effect (to account for the repeated measures in multiple brain locations).”

All analyses were rerun with more data (eyes closed resting) and 2 additional patients that have become available since the first submission. Therefore all figures and statistics have been updated throughout the paper. Other than the difference between movies and resting state which was trending before and is now significant, no results changed.